# Domain Generalizable Adaptation of 3D Vision-Language Models via Regularized Fine-Tuning

**Sneha Paul**                                      *sneha.paul@mail.concordia.ca*
*Concordia University, Canada*

**Zachary Patterson**                               *zachary.patterson@concordia.ca*
*Concordia University, Canada*

**Nizar Bouguila**                                  *nizar.bouguila@concordia.ca*
*Concordia University, Canada*

**Reviewed on OpenReview:** *https://openreview.net/forum?id=453uT7O7wc*

## Abstract

Domain adaptation remains a central challenge in 3D vision, especially for multimodal foundation models that align 3D point clouds with visual and textual data. While these models demonstrate strong general capabilities, adapting them to downstream domains with limited data often leads to overfitting and catastrophic forgetting. To address this, we introduce ReFine3D, a regularized fine-tuning framework designed for domain-generalizable tuning of 3D large multimodal models (LMMs). ReFine3D combines selective layer tuning with two targeted regularization strategies: multi-view consistency across augmented point clouds and text diversity through synonym-based prompts generated by large language models. Additionally, we incorporate point-rendered vision supervision and a test-time augmentation mechanism with confidence-based aggregation to further enhance robustness. Extensive experiments across different 3D domain generalization benchmarks show that ReFine3D improves base-to-novel class generalization by 1.36%, cross-dataset transfer by 2.43%, robustness to corruption by 1.80%, and few-shot accuracy by up to 3.11%—outperforming prior state-of-the-art methods with minimal added computational overhead.

## 1 Introduction

Point clouds are fundamental to modern 3D perception systems, powering applications from autonomous driving to augmented reality by capturing fine-grained spatial and geometric details of real-world environments. A fundamental task in these systems is recognizing 3D objects from point cloud data. Unlike structured modalities such as images or voxels, point clouds are unordered, sparse, and irregular — making them both rich in information and challenging to process. To address these challenges and build robust point cloud foundation models, recent work has adopted multimodal learning strategies (inspired by advances in vision-language models, such as CLIP (Radford et al., 2021)) by aligning 3D representations with corresponding visual and textual data (Zhang et al., 2022; Zhu et al., 2023; Zhang et al., 2023; Xue et al., 2023; 2024; Paul et al., 2026b). While these approaches show promise in capturing transferable features, adapting them to specific downstream tasks remains difficult, particularly when labelled data is scarce and has a strong distribution shift. Fine-tuning these large multimodal models (LMMs) on limited downstream data can lead to overfitting and a loss of the broad, transferable knowledge acquired during pre-training — a phenomenon known as *catastrophic forgetting* (Qi et al., 2023). This degrades the model's ability to generalize to unseen domains or corrupted data, making naive fine-tuning not only ineffective but potentially harmful in real-world, domain-shifted scenarios.

To enable efficient fine-tuning of LMMs with limited labelled data, parameter-efficient fine-tuning (PEFT) strategies such as adapter tuning (Gao et al., 2024; Song et al., 2023; Paul et al., 2026a) and prompt

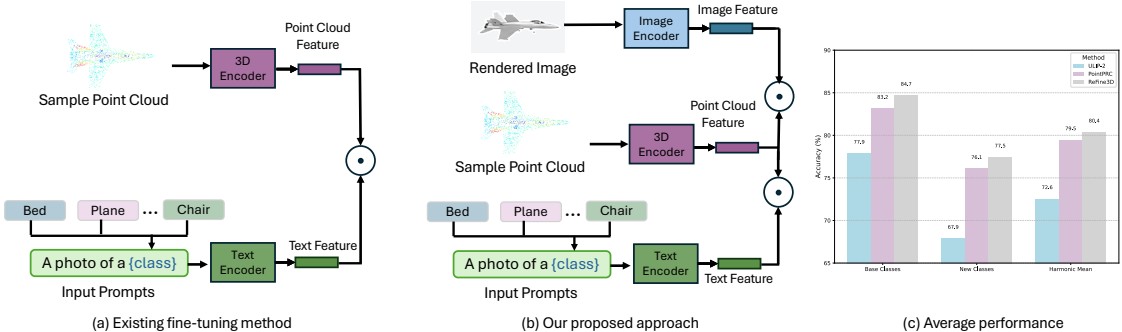

Figure 1: Unlike existing methods that discard the pre-trained vision encoder of 3D VLMs, shown in (a), our proposed fine-tuning framework utilizes the pre-trained visual priors by aligning the 3D, vision and text features in a shared representation space (presented in (b)). This strategy, along with our other proposed tuning strategies, improves the state-of-the-art in 3D domain generalizations (shown in (c)).

tuning (Khattak et al., 2023; Li et al., 2024; Zhou et al., 2022b) have been proposed. Following the success of such techniques in vision-language models, recent work in the 3D domain (e.g. PointPRC (Sun et al., 2024a)) has adopted similar PEFT strategies for fine-tuning pre-trained 3D LMMs. However, we identify several key limitations of existing PEFT approaches in the context of 3D large multi-modal models. **First**, unlike image or language encoders, existing 3D encoders typically have significantly fewer parameters. For example, PointBERT (Yu et al., 2022), a representative 3D encoder, contains 22.8 million parameters, compared to the 86 million parameter image encoder in the base version of CLIP (Radford et al., 2021). This lower capacity makes 3D encoders inherently less prone to overfitting under limited data. **Second**, point clouds differ fundamentally from images: they are unordered, unstructured, and sparse, lacking the regular grid-like organization of 2D data (Guo et al., 2021). Consequently, methods that are effective for the image domain may not necessarily perform well for the point cloud. Nonetheless, most existing PEFT strategies for point clouds naively adopt prompt tuning techniques developed for images, overlooking the distinct properties of 3D data. **Third**, state-of-the-art (SOTA) 3D prompt tuning methods such as PointPRC (Sun et al., 2024a) fine-tune the multimodal encoder by aligning point cloud features with textual representations, while discarding the image encoder during downstream adaptation. This leads to suboptimal learning, as the image encoder, pre-trained on large-scale image-text pairs, contains rich semantic knowledge that could be especially beneficial in low-resource 3D scenarios.

Motivated by these observations, in this work, we propose **Re**gularized **Fine**-tuning framework (**ReFine3D**) for tuning 3D multimodal foundation models. ReFine3D enables effective adaptation to downstream tasks while preserving the generalizable knowledge of the pre-trained models. Rather than relying on prompt tuning, ReFine3D introduces a layer-selective fine-tuning approach that strategically fine-tunes a subset of layers within the pre-trained encoder. Furthermore, to prevent overfitting due to additional layer tuning and promote generalization under low-data scenarios during fine-tuning, we incorporate two regularization strategies. The first strategy is multi-modal consistency regularization. Instead of enforcing consistency between the point cloud embedding and the class embedding (like PointPRC), we apply multiple augmentations to the input point cloud and ensure consistency across the embeddings of all augmented samples. This approach mitigates memorization and reduces overfitting to specific input instances. The second regularization is a text diversity regularization. Rather than generating text embedding directly from the class name (like PointPRC) using a hand-crafted prompt template, we first derive multiple synonyms for the class using WordNet (Miller, 1995). These synonyms are then used by a large pre-trained language model (LLM) to generate diverse descriptive sentences. We enforce consistency between the point cloud embedding and the embeddings of these sentences, promoting robustness and reducing overfitting. Finally, to retain and leverage the pre-trained vision encoder's knowledge during fine-tuning, we propose a point-rendered vision supervision method. This approach renders images from input point clouds and aligns their representations with both 3D and text embeddings, improving multi-modal consistency and performance. During inference, ReFine3D introduces a test-time augmentation mechanism with confidence-based aggregation for improving the performance. Specifically, we generate multiple augmentations of the input point cloud and apply the text diversification technique to create diverse class descriptions of the text classes. We then select the top-

*H* embeddings with the highest confidence (based on maximum softmax probability) and aggregate their predictions via majority voting. While the components in ReFine3D are not entirely novel individually, the novelty of this work lies in combining all the components into a novel fine-tuning framework for 3D LMMs to solve a very specific problem: effective domain-generalizable adaptation of 3D multimodal foundation models under limited data, while mitigating overfitting and preserving pre-trained generalization capabilities.

To evaluate our proposed framework, we conduct comprehensive experiments following the standard 3D domain generalization (3D-DG) protocol (Sun et al., 2024a), covering base-to-new generalization on five datasets (ModelNet40 (Wu et al., 2015), ShapeNetCoreV2 (Chang et al., 2015), and three ScanObjectNN variants (Uy et al., 2019)), cross-dataset generalization under four settings (OOD generalization, data corruption, domain adaptation, and sim-to-real transfer), and few-shot learning with 1, 2, 4, 8, and 16-shot settings. Our results show consistent gains across all benchmarks. Notably, ReFine3D achieves a 1.53% improvement on base classes and 1.36% on novel classes in the base-to-new setting, with a 0.92% average gain in harmonic mean across five datasets. For cross-dataset generalization, ReFine3D yields a 2.43% average improvement across five target domains. Under various types of data corruption, it outperforms the SOTA by 1.80% on average. In the few-shot setting, our method demonstrates strong generalization, achieving a 3.11% gain even with only 1-shot supervision. Furthermore, to validate the significance of each tuning strategy, we perform extensive ablation and sensitivity analysis, and provide computational cost evaluation. Overall, we make the following contributions in this paper:

- To address the challenges of existing 3D fine-tuning literature, we propose ReFine3D, a regularized fine-tuning framework that selectively fine-tunes specific layers of the pre-trained 3D VLMs for domain generalizable adaptation.

- We introduce two regularization strategies: augmentation-based consistency and text synonymization, to prevent overfitting and enhance robustness, especially under limited labelled data.

- We propose point-rendered vision supervision, which explicitly leverages the frozen CLIP image encoder's visual priors from tri-modal pre-trained models, discarded by other existing fine-tuning methods, enabling better cross-modal representation learning.

- Finally, we develop a test-time augmentation mechanism with confidence-based aggregation that aggregates predictions across multiple augmented views and textual variations during inference to boost performance.

## 2 Related Work

### 2.1 Multi-Modal Vision-Language Models in 3D point clouds

The integration of vision-language models (VLMs) into 3D point cloud analysis has revolutionized the field, enabling models to leverage both visual and textual modalities for improved generalization and open-vocabulary understanding. Pioneering works like PointCLIP (Zhang et al., 2022), pointclip2 (Zhu et al., 2023), CLIP2Point (Huang et al., 2023), ULIP (Xue et al., 2023), ULIP2 (Xue et al., 2024) extended the success of CLIP (Radford et al., 2021) to 3D by projecting point clouds into a shared embedding space with text. By aligning point cloud features with semantically rich text embeddings, these models achieved remarkable zero-shot generalization, allowing them to recognize objects from unseen categories without task-specific fine-tuning. Despite their advancements, these models share several limitations. First, they rely on full fine-tuning for downstream tasks, which is computationally expensive and risks overfitting to specific datasets (Wortsman et al., 2022). Second, while they excel in zero-shot settings, their performance degrades significantly under domain shifts, such as variations in sensor data, environmental conditions, or object appearances (Zhou et al., 2022a). For instance, models trained on synthetic datasets like ShapeNet struggle to generalize to real-world scans from ScanObjectNN or corrupted data from ModelNet-C (Ren et al., 2022). Finally, these models often lack in explicitly enforcing cross-modal consistency during fine-tuning, leading to misaligned representations that hinder generalization. These limitations highlight the need for more robust and efficient adaptation strategies, particularly in scenarios where labelled data is scarce or domain shifts are prevalent.

## 2.2 3D Domain Generalization (3D-DG)

3D domain generalization (3D-DG) aims to develop models that perform robustly across unseen domains without requiring additional fine-tuning. This is particularly challenging in real-world applications, where 3D point cloud data can vary significantly due to factors like sensor noise, occlusions, or geometric transformations. Early approaches to 3D-DG, such as PointDAN (Qin et al., 2019) and MetaSets (Huang et al., 2021), focused on domain adaptation and meta-learning to improve generalization. PointDAN introduced a multi-scale feature alignment strategy to bridge the gap between source and target domains, while MetaSets employed meta-learning on transformed point sets to handle sim-to-real geometry shifts. However, these methods were limited to small-scale datasets (e.g., ModelNet with fewer than 10,000 samples) and architectures (e.g., PointNet with 1.2M parameters), struggling to scale to the complexity of modern 3D tasks. More recent works have sought to address these limitations by leveraging large-scale pre-trained models and multi-modal learning. For instance, PDG (Wei et al., 2022b) proposed a part-level domain generalization framework, decomposing 3D objects into shared part spaces to reduce domain gaps. While effective, PDG still relies on small datasets and lacks the scalability of modern foundation models.

## 2.3 Parameter-Efficient Fine-Tuning (PEFT) for Domain Generalization (DG)

Recent literature (Tang et al., 2024; Sun et al., 2024b; Zha et al., 2023; Zhou et al., 2024b) has explored parameter-efficient fine-tuning (PEFT) techniques as a solution to the above-mentioned challenges. By introducing a small number of learnable parameters, PEFT enables efficient adaptation of pre-trained foundation models to downstream tasks while preserving their generalization capability. For instance, PPT (Sun et al., 2024b) and PointPEFT (Tang et al., 2024) demonstrated that prompt tuning can significantly improve task-specific performance without extensive retraining. These methods align point cloud features with semantically rich text embeddings, leveraging the strengths of multi-modal vision-language models like PointCLIP (Zhang et al., 2022) and ULIP (Xue et al., 2023).

Methods such as PointPEFT, IDPT, and related approaches are designed for point-cloud-only backbones trained on task-specific, in-distribution datasets (e.g., single-object classification). In contrast, our work focuses on adapting large-scale 3D vision-language models pre-trained on multimodal data, for which the objective is to preserve and improve cross-domain generalization rather than optimize performance on a single supervised task. Due to this fundamental difference in model paradigm (unimodal task-specific training vs. multimodal pre-trained vision-language models) and evaluation setting (in-distribution task performance vs. base-to-new generalization), a direct empirical comparison is not appropriate. However, existing PEFT-based approaches for 3D-DG face several limitations. First, many methods rely on multi-modal prompt tuning, which fine-tunes both point cloud and text encoders, introducing unnecessary complexity and increasing the risk of overfitting, particularly when text encoders are adapted to narrow datasets. Second, the evaluation of 3D-DG methods is hindered by the lack of diverse and robust benchmarks. While datasets like ModelNet-C (Ren et al., 2022) and PointDA (Qin et al., 2019) provide some evaluation scenarios, they often fail to capture the full spectrum of real-world domain shifts, such as cross-dataset generalization or few-shot adaptation. Furthermore, the integration of semantic diversity into prompt learning remains underexplored. Current methods (Sun et al., 2024a) primarily rely on LLM-generated or hand-crafted text descriptions, which may lack the structured semantic richness offered by external knowledge bases like WordNet (Miller, 1995).

Our work addresses these limitations by introducing a systematic fine-tuning framework for 3D multi-modal foundation models that balances task-specific adaptation with generalization. By selectively updating 3D encoder layers and incorporating consistency-based regularization across point cloud and text modalities, and utilizing the pre-trained image encoder, our work enhances robustness to domain shifts while preserving the rich priors of pre-trained vision-language models.

# 3 Methodology

## 3.1 Preliminaries

We begin by outlining the foundational components of our approach, building on vision-language and 3D multi-modal models. The Contrastive Language-Image Pre-training (CLIP) model (Radford et al., 2021)

learns aligned representations of images and text by maximizing the similarity between paired image-text embeddings while minimizing similarity for unpaired samples. Given an image embedding $\mathbf{z}_I \in \mathbb{R}^d$ and a text embedding $\mathbf{z}_T \in \mathbb{R}^d$, CLIP performs zero-shot classification by computing the similarity $\text{sim}(\mathbf{z}_I, \mathbf{z}_T) = \frac{\mathbf{z}_I \cdot \mathbf{z}_T}{\|\mathbf{z}_I\|\|\mathbf{z}_T\|}$ and selecting the class with the highest similarity to the input image embedding.

The Unified Language-augmented Point Cloud (ULIP) model (Xue et al., 2023) extends this framework to tri-modal alignment between point clouds, images, and text. ULIP processes all three modalities through separate encoders: (1) a point cloud encoder $f_P$ generating $\mathbf{z}_P \in \mathbb{R}^d$ from input point clouds $\mathbf{x} \in \mathbb{R}^{N \times 3}$, (2) an image encoder $f_I$ producing $\mathbf{z}_I \in \mathbb{R}^d$ from rendered 2D views of a point cloud, and (3) a text encoder $f_T$ yielding $\mathbf{z}_T \in \mathbb{R}^d$ from language descriptions. The model learns a unified embedding space through a symmetric contrastive loss that enforces similarity between all positive triplets $(\mathbf{z}_P, \mathbf{z}_I, \mathbf{z}_T)$ while pushing apart negative pairs. However, while ULIP's pre-trained knowledge captures generalizable 3D geometric and semantic features, it often fails to adapt to new domains or tasks with subtle geometric cues, necessitating fine-tuning for downstream tasks.

For fine-tuning, existing literature (Sun et al., 2024a) adopts a supervised approach using the cross-entropy (CE) loss. Given a point cloud sample $\mathbf{x} \in \mathbb{R}^{N \times 3}$ (where $N$ is the number of points) and its class label $y \in \{1, \ldots, C\}$, the method computes the point cloud embedding $\mathbf{z}_P = f_P(\mathbf{x})$ using the point cloud encoder $f_P$. The class prediction is obtained by comparing $\mathbf{z}_P$ with text embeddings $\{\mathbf{z}_T^c\}_{c=1}^C$ for each class, and the model is optimized using:

$$\mathcal{L}_{\text{CE}} = -\log \frac{\exp(\text{sim}(\mathbf{z}_P, \mathbf{z}_T^y)/\tau)}{\sum_{c=1}^C \exp(\text{sim}(\mathbf{z}_P, \mathbf{z}_T^c)/\tau)}, \tag{1}$$

where $\tau$ is a temperature parameter. This multi-modal fine-tuning approach preserves ULIP's cross-modal alignment while adapting to downstream tasks through discriminative learning. Despite the success of fine-tuning approaches in 3D vision-language models, we identify key limitations in existing methods, including solely relying on PEFT for tuning, discarding the pre-trained image encoder, and simply following image literature. In the following section, we discuss our proposed method, Re-Fine3D, motivated by these limitations.

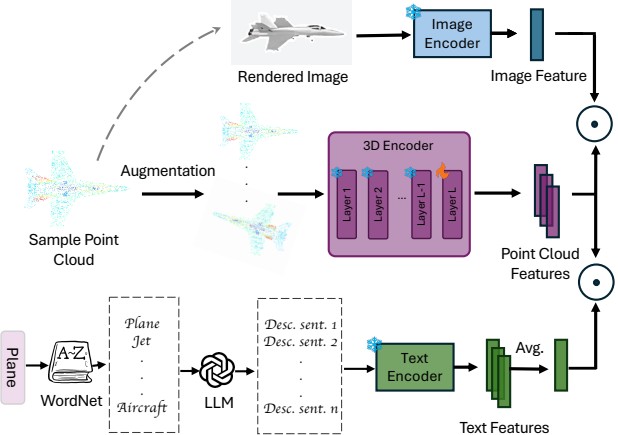

Figure 2: Our proposed Regularized Fine-tuning Framework, ReFine3D, for tuning 3D VLMs that selectively updates encoder layers while utilizing the pre-trained vision encoder's knowledge. It introduces various tuning strategies, such as layer-selective fine-tuning, augmentation- and synonym-based regularization during training, and test-time augmentation to improve task-specific learning without forgetting pre-trained task-agnostic knowledge.

### 3.2 ReFine3D

We propose **ReFine3D**, a novel framework for fine-tuning 3D LMMs for domain-generalizable adaptation while preserving pre-trained knowledge. Our approach is designed for 3D vision-language models that follow CLIP-style symmetric tri-modal contrastive pre-training, where point clouds ($\mathbf{z}_P$), images ($\mathbf{z}_I$), and text ($\mathbf{z}_T$) are mutually aligned in a shared embedding space through symmetric contrastive objectives during pre-training (Xue et al., 2023; Paul et al., 2023; 2024). This tri-modal architecture enables ReFine3D to leverage complementary supervision from both modalities during fine-tuning: the frozen image encoder $f_I$ provides rich visual priors learned from large-scale image-text pairs, while the frozen text encoder $f_T$ enables semantic guidance through diversified prompts. Unlike recent alternatives that use direct 3D-to-text alignment (e.g., Uni3D (Zhou et al., 2024a)) or asymmetric multi-task learning (e.g., OpenShape (Liu et al., 2023)), our framework requires explicit tri-modal alignment where all three encoders are available and the image/text encoders remain frozen to preserve pre-trained knowledge.

Table 1: **Complete hyperparameter configuration for ReFine3D.** All experiments use these settings unless otherwise specified.

| Component | Parameter | Value |
|---|---|---|
| **Architecture** | Backbone | ULIP-2 with PointBERT encoder |
| | 3D encoder layers ($L$) | 12 |
| | Frozen layers ($L_f$) | 11 (layers 1–11 frozen) |
| | Trainable layers | 1 (layer 12 fine-tuned) |
| | Image encoder | CLIP ViT-B/16 (frozen) |
| | Text encoder | CLIP text encoder (frozen) |
| **Training Augmentation** | Number of augmentations ($K$) | 16 |
| | Rotation range | [0°, 360°] on z-axis |
| | Scale range | [0.8, 1.2] |
| | Jitter std. deviation | 0.01 |
| **Text Diversity** | LLM for prompt generation | Qwen-2.5-7B-Instruct |
| | Number of synonyms (WordNet) | 7 per class |
| | Prompts per class ($Q$) | 7 |
| | Prompt template | "A 3D point cloud of {synonym}" |
| **Rendering Configuration** | Renderer | Blender with BlenderProc |
| | Views per object | 12 |
| | Camera distance | 1.8 meters from centroid |
| | Azimuthal angles | 0°–360° (30° increments) |
| | Elevation angles | 0°, 15°, 30° |
| | Image resolution | 512×512 pixels |
| | Focal length | 35mm |
| | Lighting | Directional (intensity 1.0) + Ambient (0.25) |
| **Loss & Optimization** | Contrastive weight ($\alpha$) | 1.0 |
| | Temperature ($\tau$) | 0.07 |
| | Optimizer | SGD |
| | Learning rate | 0.0025 |
| | Momentum | 0.99 |
| | Weight decay | 1e-5 |
| | LR schedule | Cosine annealing |
| | Training epochs | 20 |
| | Batch size | 32 |
| **Test-time Augmentation** | Test augmentations ($K_{test}$) | 5 |
| | Text prompts per class ($Q$) | 7 |
| | Top-$H$ selection | 3 |
| | Confidence metric | Maximum softmax probability |
| | Aggregation method | Majority voting |
| **Other** | Random seeds | 3 runs |
| | Point cloud sampling | 10,000 points (ModelNet40, ShapeNet) |

Unlike existing prompt-tuning approaches, ReFine3D selectively fine-tunes specific layers of the pre-trained point cloud encoder to enable downstream adaptation. Consider a transformer-based point cloud encoder $f_P$ with $L$ layers, denoted as $\{l_1, l_2, \ldots, l_L\}$. We freeze the first $L_f$ layers ($l_1$ to $l_{L_f}$) to retain pre-trained knowledge and fine-tune the last $L - L_f$ layers ($l_{L_f+1}$ to $l_L$) to capture task-specific features. Let $\theta_f$ and $\theta_t$ denote the parameters of the frozen and trainable layers, respectively. The point cloud embedding is computed as:

$$\mathbf{z}_P = f_P(\mathbf{x}; \theta_f, \theta_t). \tag{2}$$

This approach balances adaptation and generalization, as the early layers encode low-level geometric features that are broadly transferable, while the later layers capture high-level semantics that benefit from task-specific tuning.

Unlike convolutional architectures that may encode certain geometric priors, our point cloud encoder (Point-BERT, detailed in Section 3.3) is a transformer-based architecture that operates directly on raw 3D co-

Table 2: **Performance across 5 benchmarks on Base-to-new class generalization on 3D Vision Language Models (VLMs).** We adopt the performance of existing 3D VLMs with prompt tuning from (Sun et al., 2024a) and compare them with our proposed fine-tuning framework, ReFine3D. We consider ULIP and ULIP-2 as the pre-trained models since it is the most widely used models by existing fine-tuning methods. Here, Base: base class accuracy (%). New: new class accuracy (%). HM: harmonic mean of base and new class accuracy (%). Improvements are reported over existing SOTA, *PointPRC*.

(a) **Average over 5 datasets**

| Method | Base | New | HM |
|---|---|---|---|
| P-CLIP (Zhang et al., 2022) | 75.66 | 23.45 | 35.80 |
| P-CLIP2 (Zhu et al., 2023) | 74.11 | 37.84 | 50.10 |
| ULIP (Xue et al., 2023) | 77.32 | 49.01 | 59.99 |
| PointPRC (Sun et al., 2024a) | 82.19 | 61.93 | 70.64 |
| **ReFine3D** | **84.59** | **65.07** | **73.29** |
| ULIP-2 (Xue et al., 2024) | 77.91 | 67.91 | 72.57 |
| PointPRC (Sun et al., 2024a) | 83.18 | 76.10 | 79.48 |
| **ReFine3D** | **84.71** | **77.46** | **80.40** |

(b) ModelNet40

| Method | Base | New | HM |
|---|---|---|---|
| P-CLIP (Zhang et al., 2022) | 93.23 | 20.22 | 33.23 |
| P-CLIP2 (Zhu et al., 2023) | 93.98 | 45.21 | 61.05 |
| ULIP (Xue et al., 2023) | 92.80 | 50.07 | 65.05 |
| PointPRC (Sun et al., 2024a) | 95.03 | 55.27 | 69.89 |
| **ReFine3D** | **97.17** | **58.30** | **72.88** |
| ULIP-2 (Xue et al., 2024) | 91.77 | 56.47 | 69.92 |
| PointPRC (Sun et al., 2024a) | 95.30 | 64.83 | 77.17 |
| **ReFine3D** | **95.87** | **66.23** | **78.34** |

(c) S-PB_T50_RS

| Method | Base | New | HM |
|---|---|---|---|
| P-CLIP (Zhang et al., 2022) | 61.25 | 19.87 | 30.01 |
| P-CLIP2 (Zhu et al., 2023) | 56.84 | 29.92 | 39.20 |
| ULIP (Xue et al., 2023) | 56.73 | 25.80 | 35.47 |
| PointPRC (Sun et al., 2024a) | 64.20 | 49.17 | 55.69 |
| **ReFine3D** | **66.76** | **52.89** | **59.02** |
| ULIP-2 (Xue et al., 2024) | 66.40 | 66.47 | 66.43 |
| PointPRC (Sun et al., 2024a) | 73.67 | 74.27 | 73.97 |
| **ReFine3D** | **76.00** | **75.80** | **75.90** |

(d) S-OBJ_BG

| Method | Base | New | HM |
|---|---|---|---|
| P-CLIP (Zhang et al., 2022) | 72.82 | 23.00 | 34.96 |
| P-CLIP2 (Zhu et al., 2023) | 70.07 | 35.08 | 46.75 |
| ULIP (Xue et al., 2023) | 73.20 | 47.17 | 57.37 |
| PointPRC (Sun et al., 2024a) | 79.47 | 55.20 | 65.15 |
| **ReFine3D** | **82.05** | **59.77** | **69.51** |
| ULIP-2 (Xue et al., 2024) | 77.00 | 83.27 | 80.01 |
| PointPRC (Sun et al., 2024a) | 80.10 | 88.93 | 84.28 |
| **ReFine3D** | **82.35** | **90.12** | **86.06** |

(e) S-OBJ_ONLY

| Method | Base | New | HM |
|---|---|---|---|
| P-CLIP (Zhang et al., 2022) | 76.23 | 20.23 | 31.97 |
| P-CLIP2 (Zhu et al., 2023) | 71.40 | 44.39 | 54.74 |
| ULIP (Xue et al., 2023) | 74.13 | 50.80 | 60.29 |
| PointPRC (Sun et al., 2024a) | 79.23 | 65.93 | 71.97 |
| **ReFine3D** | **82.65** | **68.26** | **74.77** |
| ULIP-2 (Xue et al., 2024) | 78.60 | 76.27 | 77.42 |
| PointPRC (Sun et al., 2024a) | 83.60 | 81.10 | 82.33 |
| **ReFine3D** | **84.05** | **82.97** | **83.51** |

(f) ShapeNetCoreV2

| Method | Base | New | HM |
|---|---|---|---|
| P-CLIP (Zhang et al., 2022) | 74.78 | 33.92 | 46.61 |
| P-CLIP2 (Zhu et al., 2023) | 78.27 | 34.58 | 47.97 |
| ULIP (Xue et al., 2023) | 89.73 | 71.20 | 79.40 |
| PointPRC (Sun et al., 2024a) | 93.03 | 84.10 | 88.34 |
| **ReFine3D** | **94.82** | **86.12** | **90.26** |
| ULIP-2 (Xue et al., 2024) | 75.80 | 57.07 | 65.38 |
| PointPRC (Sun et al., 2024a) | 83.23 | 71.37 | 76.85 |
| **ReFine3D** | **85.31** | **72.62** | **78.46** |

Table 3: **Performance across 5 cross-dataset benchmarks on OOD generalization on 3D Vision Language Models (VLMs).** We report the overall accuracy (%) and standard deviation for the source and target domains, and report the average accuracy over the five target datasets in the last column.

| Method | Source ShapeNetV2 | Target ModelNet40 | S-PB_T50_RS | S-OBJ_BG | S-OBJ_ONLY | Omni3D | Avg. |
|---|---|---|---|---|---|---|---|
| P-CLIP (Zhang et al., 2022) | 67.41(0.09) | 33.20(1.86) | 15.51(0.58) | 18.59(1.40) | 22.89(2.32) | 0.48(0.17) | 22.55 |
| P-CLIP2 (Zhu et al., 2023) | 68.93(1.43) | 54.73(1.48) | 39.53(4.22) | 34.30(1.28) | 25.63(1.16) | 8.63(2.52) | 32.56 |
| ULIP (Xue et al., 2023) | 87.33(0.95) | 56.17(1.15) | 26.83(2.15) | 39.43(2.17) | 43.53(1.32) | 6.37(0.90) | 34.47 |
| PointPRC (Sun et al., 2024a) | 90.43(0.86) | 58.00(0.57) | 28.43(0.68) | 40.33(0.71) | 46.33(1.54) | 8.20(0.50) | 36.26 |
| **ReFine3D** | **91.56**(0.45) | **60.66**(0.50) | **34.03**(0.53) | **45.18**(0.60) | **50.05**(0.98) | **12.47**(2.50) | **40.48** |
| ULIP-2 (Xue et al., 2024) | 76.70(1.37) | 65.27(0.66) | 40.07(0.34) | 53.80(1.78) | 48.53(1.72) | 17.27(0.54) | 44.99 |
| PointPRC (Sun et al., 2024a) | 76.70(1.59) | 72.10(0.93) | 46.77(2.43) | 59.03(3.02) | 56.27(0.97) | 21.80(0.49) | 51.19 |
| **ReFine3D** | **80.08**(1.02) | **75.01**(0.90) | **48.93**(2.03) | **62.01**(2.21) | **58.25**(0.59) | **22.89**(0.61) | **53.62** |

ordinates without built-in invariance to geometric transformations such as rotation, scaling, or translation. Therefore, we explicitly enforce robustness to these transformations through data augmentation during training. To prevent overfitting during fine-tuning, we introduce two regularization constraints. Rather than enforcing consistency between a single point cloud embedding and its class embedding, we apply augmentation-based consistency regularization, where $K$ geometric transformations are applied to the input point cloud $\mathbf{x}$, generating augmented samples $\{\mathbf{x}_k\}_{k=1}^{K}$. Common augmentations include random rotation, scale, and jitter. Each augmented sample is passed through the encoder to obtain embeddings $\{\mathbf{z}_P^k = f_P(\mathbf{x}_k)\}_{k=1}^{K}$. We compute the CE loss for each augmented sample:

$$\mathcal{L}_{\text{CE}}^k = -\log \frac{\exp(\text{sim}(\mathbf{z}_P^k, \mathbf{z}_T^y)/\tau)}{\sum_{c=1}^{C} \exp(\text{sim}(\mathbf{z}_P^k, \mathbf{z}_T^c)/\tau)}, \tag{3}$$

and aggregate the losses as:

$$\mathcal{L}_{\text{aug}} = \frac{1}{K} \sum_{k=1}^{K} \mathcal{L}_{\text{CE}}^k. \tag{4}$$

Table 4: **Performance on corruption generalization. Here, ModelNet40 clean data is the source domain, and the target domain is ModelNet-C (Ren et al., 2022)** with different types of corruption, with the corruption severity=2.

| Method | Clean Data ModelNet | Corruption Type | | | | | | | Avg. |
|---|---|---|---|---|---|---|---|---|---|
| | | Add Global | Add Local | Drop Global | Drop Local | Rotate | Scale | Jitter | |
| P-CLIP (Zhang et al., 2022) | 80.97(1.02) | 80.97(1.02) | 80.97(1.02) | 64.95(1.08) | 68.31(1.93) | 65.75(1.19) | 72.04(1.33) | 52.09(1.28) | 69.30 |
| P-CLIP2 (Zhu et al., 2023) | 83.49(0.51) | 83.49(0.51) | 83.49(0.51) | 68.85(3.22) | 66.67(1.96) | 70.13(1.33) | 75.68(0.15) | 61.21(2.16) | 72.79 |
| ULIP (Xue et al., 2023) | 82.43(1.25) | 82.50(0.99) | 82.27(1.17) | 80.77(1.03) | 65.43(1.02) | 72.27(1.56) | 74.67(1.58) | 45.60(0.65) | 71.93 |
| PointPRC (Sun et al., 2024a) | 83.87(0.34) | 83.83(0.40) | 83.93(0.19) | 81.83(0.52) | 67.37(1.72) | 79.10(0.36) | 76.37(0.09) | 41.67(4.79) | 73.44 |
| **ReFine3D** | **84.93**(0.32) | **85.08**(0.33) | **84.53**(0.15) | **83.07**(0.41) | **70.73**(0.98) | **80.47**(0.35) | **77.73**(0.10) | 44.88(2.26) | **75.21** |
| ULIP-2 (Xue et al., 2024) | 85.07(0.21) | 81.97(0.79) | 82.03(0.96) | 79.93(0.92) | 60.03(1.21) | 80.30(0.93) | 75.77(0.74) | 44.27(2.13) | 72.04 |
| PointPRC (Sun et al., 2024a) | 86.47(0.56) | 86.57(0.48) | 86.30(0.51) | 84.87(0.48) | 67.80(1.20) | 84.60(0.22) | 81.17(1.05) | 46.43(2.45) | 76.82 |
| **ReFine3D** | **88.23**(0.50) | **88.18**(0.23) | **87.83**(0.45) | **87.01**(0.41) | **70.23**(0.98) | **86.97**(0.25) | **82.03**(0.95) | **48.08**(1.96) | **78.62** |

This encourages the model to produce consistent embeddings across augmentations, reducing memorization of specific input instances.

To reduce overfitting to class embeddings generated from class names with the text encoder, we propose a text synonymization strategy that generates diverse text prompts for each class using synonyms and a large language model (LLM). For a class $c$ with name $n_c$, we use WordNet to obtain a set of synonyms $\{s_c^1, s_c^2, \ldots, s_c^M\}$. Additionally, we leverage an LLM to generate descriptive prompt sentences incorporating $n_c$ and its synonyms, resulting in a set of prompts $\{p_c^1, p_c^2, \ldots, p_c^Q\}$. Each prompt is passed through the text encoder $f_T$ to obtain embeddings $\{\mathbf{z}_T^{c,q} = f_T(p_c^q)\}_{q=1}^Q$. Finally, the new class embedding is computed as the average:

$$\mathbf{z}_T^c = \frac{1}{Q} \sum_{q=1}^{Q} \mathbf{z}_T^{c,q}. \tag{5}$$

The averaged class embeddings are used in the CE loss (Eq. 1), promoting robustness to variations in class descriptions.

Furthermore, to leverage the knowledge embedded in the pre-trained vision encoder, we introduce *point-rendered vision supervision* to align point cloud embeddings with their rendered image embeddings. For each point cloud, we render a 2D image $\mathbf{I}$ using an off-the-shelf renderer (see details in Appendix). The image is passed through the frozen vision encoder $f_I$ to obtain an image embedding $\mathbf{z}_I = f_I(\mathbf{I})$. Here, we keep the vision encoder parameters fixed and learn only the point cloud encoder $f_P$ to align the augmented point cloud embedding $\mathbf{z}_P^k$ with the image embedding $\mathbf{z}_I$. We use a standard InfoNCE-style contrastive loss (Oord et al., 2018):

$$\mathcal{L}_{\text{contrast}}^k = -\log \frac{\exp(\text{sim}(\mathbf{z}_P^k, \mathbf{z}_I)/\tau)}{\sum_{\mathbf{z}' \in \{\mathbf{z}_I\} \cup \mathcal{Z}_{\text{neg}}} \exp(\text{sim}(\mathbf{z}_P^k, \mathbf{z}')/\tau)}, \tag{6}$$

where $\mathcal{Z}_{\text{neg}}$ includes negative image embeddings from other point clouds in the batch, and $\text{sim}(\cdot, \cdot)$ denotes cosine similarity. The contrastive losses are averaged across augmentations similar to Eq. 4:

$$\mathcal{L}_{\text{contrast}} = \frac{1}{K} \sum_{k=1}^{K} \mathcal{L}_{\text{contrast}}^k. \tag{7}$$

This pre-trained point cloud encoder is then used for fine-tuning on downstream tasks, where we align point cloud embeddings $\mathbf{z}_P$ with text embeddings $\mathbf{z}_T^y$ using cosine similarity and the CE loss (Eq. 1). Aligning point cloud features with both rendered image and text embeddings during fine-tuning allows the model to jointly leverage spatial cues from vision and semantic structure from language, leading to more discriminative and robust point cloud representations. Finally, the total loss function optimized jointly over all augmentations is:

$$\mathcal{L} = \frac{1}{K} \sum_{k=1}^{K} \left( \mathcal{L}_{\text{CE}}^k + \alpha \cdot \mathcal{L}_{\text{contrast}}^k \right), \tag{8}$$

where $\alpha$ is a hyperparameter to balance the contributions of the contrastive and classification losses.

To enhance generalization at inference, we introduce a test-time augmentation mechanism with confidence-based aggregation inspired by leveraging additional computation at inference (Wei et al., 2022a; Turpin

Table 5: **Performance on cross-dataset generalization on _PointDA_**(Qin et al., 2019). M: ModelNet, S: ShapeNet, S*: ScanNet. We present the average across 6 evaluation setups in the last column.

| Method | M → S | M → S* | S → M | S → S* | S* → M | S* → S | **Avg.** |
|---|---|---|---|---|---|---|---|
| P-DAN (Qin et al., 2019) | 64.2 | 33.0 | 47.6 | 33.9 | 49.1 | 64.1 | 48.7 |
| MetaSets (Huang et al., 2021) | 86.0 | 52.3 | 67.3 | 42.1 | 69.8 | 69.5 | 64.5 |
| PDG (Wei et al., 2022b) | 85.6 | 57.9 | 73.1 | 50.0 | 70.3 | 66.3 | 67.2 |
| I-OODG (Zhang et al., 2024) | 83.7 | 56.4 | 71.7 | 57.6 | 69.5 | 73.5 | 67.8 |
| P-CLIP2 (Zhu et al., 2023) | 40.53 | 26.40 | 31.33 | 35.57 | 16.30 | 24.97 | 29.18 |
| ULIP (Xue et al., 2023) | 74.33 | 38.23 | 35.17 | 36.17 | 24.70 | 60.67 | 44.88 |
| ULIP-2 (Xue et al., 2024) | 84.80(2.69) | 48.10(2.13) | 83.20(4.17) | 42.00(4.18) | 60.43(4.83) | 70.50(6.22) | 64.84 |
| PointPRC(Sun et al., 2024a) | 89.00(1.18) | 51.37(**1.03**) | 89.87(2.38) | 49.57(2.50) | 85.57(3.80) | 83.07(4.21) | 74.74 |
| **ReFine3D** | **90.22(1.05)** | **53.17**(1.11) | **92.07(2.02)** | **53.25(1.39)** | **87.31(2.23)** | **87.16(3.08)** | **77.19** |

Table 6: **Performance on cross-dataset generalization on _Sim-to-Real_**(Huang et al., 2021) in two evaluation settings, MN_11 → SONN_11, SN_9 → SONN_9.

| Method | MN_11 → SONN_11 | | | SN_9 → SONN_9 | | | **Avg.** |
|---|---|---|---|---|---|---|---|
| | OBJ | OBJ_BG | PB_T50_RS | OBJ | OBJ_BG | PB_T50_RS | |
| MetaSets-P (Huang et al., 2021) | 60.3 | 52.4 | 47.4 | 51.8 | 44.3 | 45.6 | 50.3 |
| MetaSets-D (Huang et al., 2021) | 58.4 | 59.3 | 48.3 | 49.8 | 47.4 | 42.7 | 51.0 |
| PDG-P (Wei et al., 2022b) | 67.6 | 58.5 | 56.6 | 57.3 | 51.3 | 51.3 | 57.1 |
| PDG-D (Wei et al., 2022b) | 65.3 | 65.4 | 55.2 | 59.1 | 59.3 | 51.0 | 59.2 |
| P-CLIP2 (Zhu et al., 2023) | 18.67 | 15.57 | 15.63 | 53.00 | 47.83 | 35.83 | 31.09 |
| ULIP (Xue et al., 2023) | 21.60 | 18.03 | 13.63 | 54.83 | 54.17 | 40.87 | 33.86 |
| ULIP-2 (Xue et al., 2024) | 62.73(0.95) | 68.23(0.86) | 52.83(1.10) | 66.90(2.77) | 70.50(2.48) | 54.03(2.75) | 62.54 |
| PointPRC (Sun et al., 2024a) | 68.43(1.07) | 69.47(0.95) | 55.30(**2.00**) | 65.83(1.35) | 72.53(**0.47**) | 58.57(1.17) | 65.02 |
| **ReFine3D** | **70.11(0.91)** | **72.53(0.78)** | **56.87(2.15)** | **66.92(1.22)** | **73.03(0.62)** | **60.00(0.52)** | **66.58** |

et al., 2023). Unlike training-free test-time adaptation methods (Sun et al., 2025) that operate in zero-shot settings, our approach leverages the task-specific knowledge acquired during few-shot fine-tuning. For a test point cloud $\mathbf{x}_{\text{test}}$, we generate $K_{\text{test}}$ augmentations $\{\mathbf{x}_{\text{test}}^k\}_{k=1}^{K_{\text{test}}}$ and compute their embeddings $\{\mathbf{z}_P^k = f_P(\mathbf{x}_{\text{test}}^k)\}_{k=1}^{K_{\text{test}}}$. On the text branch, we use the synonymization strategy to generate multiple class embeddings $\{\mathbf{z}_T^{c,q}\}_{q=1}^Q$ per class and aggregate results via averaging. Then we compute similarities of these text embeddings with the augmented point cloud embeddings. For each class, we select the top-$H$ embeddings with the highest confidence (based on maximum softmax probability) and aggregate their predictions via majority voting. This approach leverages additional computation to improve robustness and accuracy by reducing erroneous predictions during inference.

### 3.3 Implementation Details

We adopt ULIP-2 (Xue et al., 2024) as the pre-trained backbone, since it is the most widely adopted pre-trained foundation model in the existing fine-tuning literature. We perform all experiments using three different random seeds and report the mean and standard deviation to ensure statistical reliability. For model training, we use stochastic gradient descent (SGD) as the optimizer with an initial learning rate of 0.0025. We present a detailed hyperparameter configuration in Table 1.

### 3.4 Details on Image Rendered

Rendering images from point clouds is a one-time offline operation independent of the actual training process. Although we generated the rendered images for our method, many existing datasets already provide rendered images alongside 3D data (e.g., widely used 3D-vision benchmarks). In our implementation, rendering has a throughput of approximately 100 samples per second using Blender and BlenderProc (Denninger et al., 2019). ModelNet40 and ShapeNetCoreV2 sampled to 10,000 points via uniform sampling to align with standard practices (Qi et al., 2017), while ScanObjectNN and Objaverse retain native point densities (2,000–50,000 points) to preserve real-world noise and complex geometries, incorporating surface normals and RGB attributes when available. The rendering configuration is optimized for consistency: a directional light (intensity 1.0, 45° elevation, 30° azimuth) simulates natural sunlight, paired with an ambient light (intensity 0.25) to soften shadows. The camera employs a perspective projection with a 35mm focal length, 32mm sensor width, and 512x512 pixel resolution, with extrinsics placing the camera 1.8 meters from the

object's centroid across 12 viewpoints (azimuthal angles 0°–360° in 30° increments, elevation angles 0°, 15°, 30°). Materials use a Lambertian diffuse shader (reflection coefficient 0.8) based on point cloud normals or RGB data, ensuring realistic shading for synthetic and real-world objects without specular highlights. A solid gray background (RGB 0.5, 0.5, 0.5) maintains focus on objects, aligning with common practices in ShapeNetCoreV2 and ModelNet40.

# 4 Results and Experiments

## 4.1 Evaluation Protocol

To ensure fair comparison with existing approaches in 3D domain generalization (3D-DG), we adopt the evaluation protocol established by prior work (Sun et al., 2024a). This protocol includes three key benchmarks: base-to-new generalization, cross-dataset generalization, and few-shot learning. The base-to-new benchmark is conducted on five widely used datasets—ModelNet40 (Wu et al., 2015), ShapeNetCoreV2 (Chang et al., 2015), and three variants of ScanObjectNN (Uy et al., 2019) (S-PB_T50_RS, S-OBJ_BG, and S-OBJ_ONLY). Each dataset is evenly divided into base and novel classes. Fine-tuning is performed on the base classes, while model generalization is evaluated on novel classes. For cross-dataset generalization, we consider four settings: (i) out-of-distribution (OOD) generalization, where models trained on one domain are evaluated on different domains; (ii) data corruption, where training is conducted on clean ModelNet40 and testing is done on its corrupted counterpart, ModelNet-C (Ren et al., 2022); (iii) domain adaptation following PointDA (Qin et al., 2019); and (iv) sim-to-real transfer using the protocol introduced in (Huang et al., 2021). Additionally, to assess performance

Table 7: **Performance across 5 benchmarks on few-shot generalization.** We report the overall accuracy and the average on the different shots ranging from 1,2,4,8, and 16.

| Setting | Methods | Average | ModelNet40 | S-PB_T50_RS | S-OBJ_BG | S-OBJ_ONLY | Omni3D |
|---|---|---|---|---|---|---|---|
| 1-shot | PointPRC | 63.49 | 65.53 | 52.10 | 64.50 | 64.80 | 70.50 |
| | ReFine3D | 66.60 | 67.87 | 55.78 | 67.48 | 69.57 | 72.29 |
| 2-shot | PointPRC | 66.88 | 68.41 | 53.50 | 72.00 | 69.50 | 71.00 |
| | ReFine3D | 68.18 | 71.13 | 55.32 | 72.18 | 69.84 | 72.43 |
| 4-shot | PointPRC | 72.21 | 70.05 | 68.50 | 73.50 | 75.00 | 74.00 |
| | ReFine3D | 75.14 | 73.64 | 70.96 | 76.44 | 76.16 | 78.51 |
| 8-shot | PointPRC | 72.77 | 72.34 | 63.50 | 76.00 | 78.00 | 74.00 |
| | ReFine3D | 75.82 | 75.02 | 68.15 | 77.91 | 77.93 | 80.08 |
| 16-shot | PointPRC | 78.92 | 77.17 | 73.97 | 84.28 | 82.33 | 76.85 |
| | ReFine3D | 80.44 | 78.34 | 75.90 | 86.28 | 83.51 | 78.19 |

under limited supervision, we conduct few-shot learning experiments with 1, 2, 4, 8, and 16 labelled samples per class from the base set, and evaluate on the full test set. If not specified, we perform the experiments on 16-shot setting.

## 4.2 Base-to-new Class Generalization

We evaluate our proposed method on the base-to-new class generalization setting across five 3D benchmarks, as summarized in Table 2. The results demonstrate that ReFine3D consistently outperforms existing methods across all metrics, including base accuracy, new class accuracy, and their harmonic mean (HM). Specifically, ReFine3D improves the generalization performance on both novel and base classes, achieving the highest new class accuracy of 77.46% and base class accuracy of 84.71% on average.

Notably, we observe the largest improvement in HM on datasets with higher variability and more challenging semantic gaps between base and novel classes, such as S-PB_T50_RS with HM gains of 1.93% over PointPRC (Sun et al., 2024a). ReFine3D also shows strong and consistent performance on ModelNet40 and S-OBJ_BG and ShapeNetCoreV2, confirming the method's robustness across diverse scenarios. On average across the five datasets, ReFine3D achieves an HM of 80.40%, representing a relative gain of 0.92% over the previous state-of-the-art PointPRC.

Table 8: **Pairwise comparison between ReFine3D and PointPRC.** We report the percentage of samples where both methods are correct, only ReFine3D is correct, only PointPRC is correct, or both methods are wrong. Win Rate is computed as $\frac{\text{ReFine3D Only}}{\text{ReFine3D Only+PointPRC Only}}$.

| Dataset | Both Correct | ReFine3D Only | PointPRC Only | Both Wrong | Win Rate |
|---|---|---|---|---|---|
| ModelNet40 | 63.2 | 15.1 | 11.1 | 10.6 | **57.6** |
| S-PB-T50-RS | 51.8 | 24.0 | 22.2 | 2.0 | **51.9** |
| S-OBJ-BG | 52.1 | 24.8 | 23.1 | 0.0 | **51.8** |
| S-OBJ-ONLY | 65.7 | 17.6 | 16.7 | 0.0 | **51.3** |
| ShapeNetV2 | 70.5 | 15.9 | 13.6 | 0.0 | **53.9** |
| **Average** | **60.7** | **19.5** | **17.3** | **2.5** | **53.3** |

## 4.3 Cross-Dataset Generalization

In this section, we analyze our proposed method's performance for OOD generalization and data corruption settings. We present a detailed analysis of the other two settings, domain adaptation (Qin et al., 2019) and sim-to-real (Huang et al., 2021).

**OOD generalization.** Out-of-distribution (OOD) generalization assesses a model's capacity to transfer knowledge learned from a known domain to unseen target domains. In this benchmark, we use ShapeNetV2 as the source domain and evaluate the model's generalization on five diverse target datasets. As shown in Table 3, ReFine3D significantly outperforms prior 3D vision-language models in both source and target domains. Our method improves existing SOTA by +3.38% on the source domain with an average of +2.43% across 5 target domains.

Table 9: **McNemar's test on paired disagreements.**

| Dataset | p-value |
|---|---|
| ModelNet40 | **0.031** |
| S-PB-T50-RS | **0.042** |
| S-OBJ-BG | **0.038** |
| S-OBJ-ONLY | **0.047** |
| ShapeNetV2 | **0.029** |

**Data corruption.** In the real world, point clouds often suffer from data corruption due to sensor noise, incomplete scans, and geometric irregularities. To evaluate the robustness of our framework under such realistic perturbations, we test on ModelNet-C (Ren et al., 2022), which applies seven common corruptions at severity level 2. As shown in Table 4, ReFine3D achieves the highest accuracy on both clean and corrupted data settings. On the clean ModelNet40 dataset, ReFine3D improves SOTA by +1.76%. When averaged across 7 corruptions, ReFine3D yields a substantial +1.80% gain, consistently leading in all corruption types.

**Domain adaptation.** PointDA, a 3D domain adaptation benchmark pioneered by PointDAN (Qin et al., 2019), features six evaluation scenarios outlined in Table 5. Unlike earlier techniques such as MetaSets (Huang et al., 2021), PDG (Wei et al., 2022b), and I-OODG (Zhang et al., 2024), which utilize the complete training dataset across each scenario, our ReFine3D method relies on a few-shot learning approach with 16 samples. The outcomes underscore ReFine3D's strong performance in domain adaptation, delivering an average accuracy of 77.19% across all setups. Remarkably, it outpaces the existing SOTA method, PointPRC, by 2.45%, with notable gains including a 4.09% boost when the source domain is ScanNet and target domain is

Table 10: **Ablation study on the different fine-tuning strategies of our proposed framework.** We report the overall accuracy and harmonic mean (HM) on the Base-to-new generalization setting on S-PB_T50_RS.

| Method | Base | New | HM |
|---|---|---|---|
| ReFine3D (ours) | 76.00 | 75.80 | 75.90 |
| w/o vision guidance | 73.98 | 74.50 | 74.24 |
| w/o test-time augmentation | 74.06 | 74.93 | 74.50 |
| w/o layer-specific fine-tuning | 74.12 | 74.58 | 74.35 |
| w/o training augmentation | 74.25 | 74.62 | 74.43 |
| w/o text synonymization | 74.18 | 74.06 | 74.12 |

ShapeNet, and a 3.68% rise on ShapeNet to ScanNet generalization, highlighting its effectiveness in improving cross-dataset generalization with limited data.

**Sim-to-Real generalization.** The Sim-to-Real evaluation assesses cross-domain generalization by transitioning from simulated to real-world data, a concept initially explored by MetaSets (Huang et al., 2021) and

later expanded by PDG (Wei et al., 2022b). In this evaluation, ModelNet (Wu et al., 2015) and ShapeNet (Chang et al., 2015) serve as synthetic point cloud sources, while ScanObjectNN (Uy et al., 2019) is derived from real-scanned data. Unlike MetaSets and PDG, which rely on the entire training set in the source domain for supervised learning, our approach utilizes only 16-shot prompt tuning. As shown in Table 6, our framework consistently boosts generalization across various 3D models, with an improvement of 1.56% over the SOTA method averaged over six datasets.

### 4.4 Few-shot Generalization

To assess the effectiveness of ReFine3D in label-constrained scenarios, we evaluate its performance across five benchmarks under varying few-shot settings. As shown in Table 7, ReFine3D consistently outperforms existing SOTA across all shots and benchmarks. The performance gains are especially notable in extremely low-shot scenarios. For instance, ReFine3D achieves an average improvement of 3.11% in the 1-shot setting. Notably, performance continues to improve steadily with more labelled samples, reaching 80.44% at 16 shots. This trend underscores the robustness and adaptability of our approach as label availability increases.

### 4.5 Sample-level Analysis

We report a paired, sample-level disagreement analysis between ReFine3D and PointPRC (Table 8). Rather than relying solely on aggregate accuracy, we have analyzed predictions on a per-sample basis and partitioned the test set into four mutually exclusive categories: (1) both correct, (2) ReFine3D only correct, (3) PointPRC only correct, and (4) both wrong. This paired evaluation isolates the exact samples on which the methods disagree and directly measures relative superiority. Across all five benchmarks, ReFine3D wins more disagreements than it loses. On average, ReFine3D correctly classifies 19.5% of samples that PointPRC misclassified, compared to 17.3% vice versa, yielding an average win rate of 53.3%. Importantly, ReFine3D achieves a win rate greater than 50% on every dataset, demonstrating consistent per-sample advantage rather than isolated gains.

Table 11: **Comprehensive ablation of all component combinations in ReFine3D.** PA = point cloud augmentation, TA = test-time augmentation, TS = text synonymization, VG = vision guidance. We report harmonic mean (HM) on Base-to-New generalization of S-PB-T50-RS.

| PA | TA | TS | VG | HM | Label |
|----|----|----|----|-------|--------|
| ✗ | ✗ | ✗ | ✗ | 71.67 | All Off |
| ✓ | ✗ | ✗ | ✗ | 72.63 | Only PA |
| ✗ | ✓ | ✗ | ✗ | 72.75 | Only TA |
| ✗ | ✗ | ✓ | ✗ | 72.77 | Only TS |
| ✗ | ✗ | ✗ | ✓ | 72.97 | Only VG |
| ✓ | ✓ | ✗ | ✗ | 73.50 | PA+TA |
| ✓ | ✗ | ✓ | ✗ | 73.59 | PA+TS |
| ✓ | ✗ | ✗ | ✓ | 73.42 | PA+VG |
| ✗ | ✓ | ✓ | ✗ | 73.62 | TA+TS |
| ✗ | ✓ | ✗ | ✓ | 73.52 | TA+VG |
| ✗ | ✗ | ✓ | ✓ | 73.44 | TS+VG |
| ✓ | ✓ | ✓ | ✗ | 74.43 | w/o PA |
| ✓ | ✓ | ✗ | ✓ | 74.35 | w/o TS |
| ✓ | ✗ | ✓ | ✓ | 74.50 | w/o TA |
| ✗ | ✓ | ✓ | ✓ | 74.24 | w/o VG |
| ✓ | ✓ | ✓ | ✓ | 75.90 | **ReFine3D** |

To formally assess whether these disagreement asymmetries are statistically significant, we performed McNemar's test on the paired disagreement counts (ReFine3D Only vs. PointPRC Only). The reported p-values evaluate whether the difference in prediction outcomes between the two methods is statistically significant. Lower p-values indicate stronger evidence that the observed performance difference is not due to chance. As shown in the Table 9, all datasets yield p-values below 0.05, indicating that the observed asymmetries are unlikely to arise from random variation.

Table 12: **Ablation study on tuning different layers of point cloud encoder.** We report the overall accuracy and harmonic mean (HM) on the Base-to-new generalization setting on S-PB_T50_RS.

| Number of layers | Base | New | HM |
|-------------------|-------|-------|-------|
| Last layer attention block only | 76.00 | 75.80 | 75.90 |
| Last layers (last 4 attention blocks) | 75.82 | 75.14 | 75.48 |
| Middle layers (3-7 attention block) | 74.96 | 74.34 | 74.65 |
| Early layers (First 4 attention blocks) | 73.15 | 74.02 | 73.58 |
| Full finetuning | 73.06 | 70.57 | 71.79 |

### 4.6 Ablation and Sensitivity Study

**Different fine-tuning strategies of ReFine3D.** We present an ablation study on the different fine-tuning strategies of our proposed method, ReFine3D, in Table 10. ReFine3D achieves the highest performance with a harmonic mean (HM) of 75.90% when all proposed tuning strategies are applied. Removing each component individually results in a consistent drop in performance, highlighting their collective sig-

nificance. Notably, disabling vision guidance leads to the most significant decline, with HM dropping by 1.66%. Text synonymization also contributes substantially, with its removal reducing HM by 1.78%. Other tuning strategies—test-time scaling, layer-specific fine-tuning, and training augmentation—each contributes approximately 1.4–1.5% to the overall performance when removed.

Table 11 demonstrate that the four fine-tuning strategies interact synergistically. Each component individually provides consistent improvement, yet when combined, their contributions compound substantially. Specifically, ReFine3D achieves 4.23% overall improvement over the baseline, which exceeds both the sum of individual effects and any pairwise combination, indicating strong synergy among all components.

**Number of fine-tuned layers**. Table 12 presents an ablation study on the layer-selective fine-tuning strategy. In this study, we keep the vision and text encoder frozen while selectively tuning different layers of the 3D encoder (e.g. PointBERT). We tune the layers at different levels, such as early, mid and last layers. The table shows that tuning the last layers results in the highest performance gain, indicating the significance of high-level semantic adaptation for generalization. On the contrary, the early and mid layers encode general low-level geometric patterns, thus contributing less to transferring knowledge in downstream tasks.

Table 13: **Ablation of test-time inference strategies on S-PB T50 RS.** Here, TTA represents test-time augmentation.

| Test-Time Strategy | Base | New | HM |
|---|---|---|---|
| Single view (no scaling) | 74.12 | 74.58 | 74.35 |
| Simple TTA (avg. 5 views) | 74.80 | 75.02 | 74.91 |
| TTA + text diversity | 75.33 | 75.48 | 75.40 |
| Full Test-time augmentation (ours) | **76.00** | **75.80** | **75.90** |

**Test-time inference strategies**. We analyze the effectiveness of our test-time augmentation mechanism by comparing different inference strategies. Table 13 shows that simple test-time augmentation (averaging predictions over 5 views) provides modest gains (+0.56% HM), while adding text diversity improves further (+1.05% HM). Our full test-time augmentation, which combines augmentation, text diversity, and top-$H$ confidence-based selection, achieves the best performance (+1.55% HM over single-view inference). Note that our test-time augmentation addresses a different problem than training-free test-time adaptation methods like Point-Cache (Sun et al., 2025), which operate without any labelled downstream data. Point-Cache achieves robustness through dynamic feature caching in pure zero-shot settings, while our approach leverages task-specific knowledge from few-shot fine-tuning.

Table 14: **Ablation study on text prompts generated from different sources.** We report the overall accuracy and harmonic mean (HM) on the Base-to-new generalization setting on S-PB_T50_RS. Here, GPT-3.5 represents GPT-3.5-turbo.

| Text propmts | Base | New | HM |
|---|---|---|---|
| Manual | 73.98 | 74.50 | 74.24 |
| Qwen-2.5 | **76.00** | **75.80** | **75.90** |
| LLaMa-3 | 74.06 | 74.93 | 74.50 |
| Mistral | 74.12 | 74.58 | 74.35 |

**Text prompts from various sources.** Table 14 presents an ablation study comparing text prompts generated from different sources—Manual, Qwen-2.5, LLaMa-3, and Mistral—on a Base-to-New generalization task. Among these, Qwen-2.5 prompts deliver the highest results across all metrics (Base: 76.00, New: 75.80, HM: 75.90), clearly outperforming manually crafted prompts. This suggests that high-capacity language models can generate richer or more relevant textual cues. When compared to other LLMs, Qwen-2.5 also surpasses LLaMa-3 and Mistral, which still improve over manual prompts. Overall, these findings demonstrate that leveraging advanced LLM-generated prompts can significantly enhance the generalization capability of vision-language models on both seen and unseen categories.

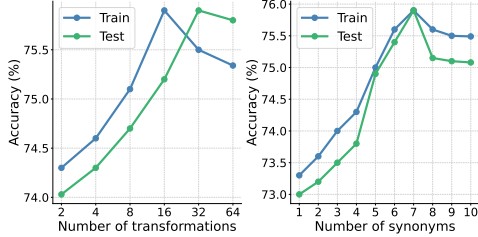

Figure 3: The impact of transformation strength and number of text synonyms during train and test time.

Table 15: **Qualitative examples of WordNet synonyms and LLM-generated prompts.** For each class, we show WordNet synonyms and corresponding LLM-generated descriptive prompts.

| Class | WordNet Synonyms | LLM-Generated Prompts |
|---|---|---|
| Airplane | aircraft, plane, jet | A 3D point cloud of an aircraft with wings and a fuselage
A 3D point cloud of a plane designed for air travel
A 3D point cloud of a jet with engines mounted on the body |
| Chair | seat, furniture, seating | A 3D point cloud of a seat with a backrest and legs
A 3D point cloud of furniture designed for sitting
A 3D point cloud of seating with armrests and a base |
| Lamp | light, lighting fixture, luminaire | A 3D point cloud of a light source with a shade and stand
A 3D point cloud of a lighting fixture for illumination
A 3D point cloud of a luminaire with electrical components |

To provide transparency in our text diversification pipeline, we include qualitative examples of WordNet synonyms and LLM-generated prompts used in ReFine3D. Table 15 shows examples from three representative classes with their WordNet synonyms and corresponding LLM-generated descriptive prompts using Qwen-2.5-7B-Instruct. These examples demonstrate the semantic diversity captured through our synonymization strategy, which helps the model learn more robust and generalizable representations by avoiding overfitting to specific class name phrasings.

**Transformation strength.** Figure 3 *(left)* presents an analysis of the ReFine3D's performance during training and test time under various transformation settings. The left plot displays model performance under different numbers of geometric augmentations applied to point clouds. During training, accuracy peaks when 16 geometric augmentations are applied to the point cloud. During test time, the highest performance is achieved when 32 augmentations are applied.

Figure 3 *(right)* presents the model's performance with varying numbers of synonyms used in the text

Table 16: **Sensitivity study on different numbers of rendered images.** We report the overall accuracy and harmonic mean (HM) on the Base-to-new generalization setting on S-PB_T50_RS.

| Number of views | Base | New | HM |
|---|---|---|---|
| 1 | 76.00 | **75.80** | 75.90 |
| 2 | 77.46 | 75.68 | 76.56 |
| 3 | 79.23 | 74.10 | **76.57** |
| 4 | 80.10 | 72.00 | 75.78 |
| 5 | **80.60** | 70.30 | 74.72 |

prompts. At both train and test time, 7 synonyms per word applied to the text prompts give the highest performance, indicating that the severity of transformations is critical to the model's performance.

**Vision Supervision Trade-off Analysis.** Table 16 presents an ablation study evaluating the number of rendered image views affecting model performance under the Base-to-New generalization setting. Initially, increasing the number of views from 1 to 3 improves base class accuracy (from 76.00 to 79.23) by providing richer visual information. However, novel class accuracy declines, reflecting reduced generalization. As the number of views increases further (to 4 and 5), base accuracy continues to rise due to stronger memorization of base examples (reaching 80.60 at 5 views), but novel accuracy drops more noticeably (to 70.30 at 5 views), suggesting view-specific overfitting.

This counterintuitive trade-off arises from view-specific overfitting. With multiple rendered views, the model learns to align point cloud features with view-dependent visual patterns (specific lighting, camera angles, surface normals) that are abundant in base class training data but absent from novel classes. Unlike geometric augmentations, which preserve object identity, multiple rendered views create consistent but view-specific visual anchors that the model memorizes rather than learning view-invariant 3D representations. Through our contrastive vision supervision loss, using a single view forces the encoder to extract the most salient and transferable 3D-to-2D mappings, whereas multiple views allow the encoder to overfit by learning all view-specific patterns simultaneously.

Table 17: **Sensitivity study on the loss balancing hyperparameter $\alpha$.** We report accuracy and harmonic mean (HM) under the Base-to-New generalization setting.

| $\alpha$ | Base | New | HM |
|---|---|---|---|
| 0.5 | 75.10 | 74.20 | 74.65 |
| 1.0 | **76.00** | **75.80** | **75.90** |
| 1.5 | 75.80 | 74.00 | 74.89 |

Our choice of 1 rendered view is an informed design decision that achieves three objectives: (1) balanced generalization with minimal base-novel gap (0.20% vs. 10.30% at 5 views), (2) sufficient visual supervision—removing vision guidance drops HM by 1.66%, the largest single-component penalty, and (3) computational efficiency. This finding validates that vision supervision quality matters more than quantity, and strategic constraint in multimodal alignment is essential for maintaining generalization under domain shift.

**Loss balancing hyperparameter, ($\alpha$).** Table 17 presents an ablation study on the loss balancing hyperparameter, $\alpha$, which controls the relative contribution of the contrastive loss in the final objective function. The results show that equal importance to both loss components optimally supports balanced learning and transferability and yields the best overall performance. Lower or higher values result in decreased HM scores, suggesting that under- or over-weighting the contrastive loss degrades generalization.

Table 18: **Comparison between different LoRA configurations.** Results are reported in terms of harmonic mean (HM).

| Method | HM |
|---|---|
| PointPRC | 73.97 |
| LoRA (Rank 4) | 71.26 |
| LoRA (Rank 8) | 74.00 |
| LoRA (Rank 12) | 73.03 |
| LoRA (Attention Only) | 72.82 |
| LoRA (MLP Only) | 74.00 |
| LoRA (Attn + MLP) | 71.08 |
| **ReFine3D** | **75.90** |

**LoRA baselines.** To validate our layer-selective strategy against standard PEFT alternatives, we compare ReFine3D with LoRA (Hu et al., 2022) under varying rank configurations and insertion locations (Table 18). LoRA performance varies considerably: while Rank 8 and MLP-only match PointPRC (74.00), other configurations underperform it, with Attn+MLP dropping to 71.08—reflecting the risk of disrupting pretrained geometric representations in early layers critical for cross-domain transfer.

## 4.7 Computational Complexity

In Table 19, we examine the training efficiency of our proposed method, ReFine3D. All methods are trained for 20 epochs using *NVIDIA V100 GPU*. On both datasets, ReFine3D shows a marginal increase in training time. This is due to tuning additional layers. However, the cost remains competitive and well within practical limits compared to the existing SOTA. Since ReFine3D does not add any extra parameters (e.g. prompts, adapters, like existing methods), the computational cost and memory footprint at inference remain the same as the pre-trained encoder.

Table 19: **Run time analysis** during training on ModelNet40 and S-PB_T50_RS variant of ScanObjectNN dataset. We report the time count in *seconds*.

| Method | Dataset | |
|---|---|---|
| | MN40 | S-PB_T50_RS |
| ULIP-2 | 132 | 106 |
| PointPRC | 159 | 112 |
| **Refine3D** | 162 | 115 |

## 4.8 Scalability Analysis

We further evaluate our framework on the larger and more challenging Objaverse-LVIS (Deitke et al., 2023) dataset, which serves as the target domain in this experiment. Objaverse-LVIS is a curated subset of the recently released Objaverse, containing 46,205 point clouds across 1,156 classes, including only a single instance for some classes, making it particularly difficult for conventional point cloud recognition models.

Table 20: **Analysis on scalability**. Here, we use a smaller dataset ShapeNetV2, as the source domain and Objaverse-LVIS as the target domain.

| Method | Source | Target |
|---|---|---|
| | ShapeNetV2 | Objaverse-LVIS |
| ULIP-2 | 76.70(1.37) | 14.80(0.22) |
| PointPRC | 76.70(1.59) | 18.07(0.49) |
| **ReFine3D** | **78.50** (1.20) | **21.43**(0.41) |
| $\Delta$ | ↑ 1.80 | ↑ 3.36 |

Table 21: **Test-time augmentation trade-offs** on S-PB_T50_RS. We report accuracy (HM), inference latency per sample, and throughput. All measurements on NVIDIA V100 GPU. Here, $K_{test}$ represents number of augmented point clouds and $Q$ represents number of text prompts.

| Configuration | $K_{test}$ | $Q$ | **HM** (%) | **Latency (ms)** |
|---|---|---|---|---|
| Single view (baseline) | 1 | 1 | 74.35 | 45 |
| TTA (5 views) | 5 | 1 | 74.91 | 49 |
| Text diversity | 1 | 7 | 74.68 | 46 |
| TTA + text diversity (ours) | 5 | 7 | 75.90 | 50 |

As shown in Table 20, ReFine3D achieves substantial improvements, a +1.80% gain on the source and a significant +3.36% gain on the extensive target dataset.

### 4.9 Test-time Augmentation Trade-offs

Our test-time augmentation mechanism improves performance with minimal computational overhead. Table 21 shows the accuracy-latency trade-off on S-PB_T50_RS using an NVIDIA V100 GPU. Single-view inference achieves 74.35% HM with 45ms latency. Adding test-time augmentation ($K_{test} = 5$) improves accuracy to 74.91% (+0.56%) with only 49ms latency. Text diversity ($Q = 7$ prompts) provides 74.68% accuracy with 46ms latency, as precomputed text embeddings add virtually no overhead. Our full configuration ($K_{test} = 5$, $Q = 7$) achieves 75.90% HM with 50ms latency—a +1.55% accuracy gain for only 5ms additional cost. The modest overhead stems from encoding augmented point clouds through the 3D encoder, while text diversity introduces negligible cost since embeddings are precomputed and cached.

### 4.10 Discussion

Our work provides several incremental insights that advance understanding of 3D multimodal fine-tuning:

**(1) Modality-Aware Regularization Synergy.** Table 11 shows that augmentation-based regularization alone provides limited gains, but when combined with vision-guided multimodal alignment, contributions compound substantially to achieve 4.23% total improvement. This demonstrates that integrating consistency regularization across point clouds with multimodal alignment is essential for learning generalizable representations under domain shift.

**(2) Quality vs. Quantity in Vision Supervision.** Table 16 reveals that increasing rendered views paradoxically degrades novel class accuracy (from 75.80% to 70.30%), indicating view-specific overfitting. This counter-intuitive finding shows that strategic constraint in multimodal alignment—using minimal yet sufficient supervision—is more important than quantity for preserving generalization.

**(3) Asymmetric Encoder Tuning.** Table 12 shows layer-selective fine-tuning achieves 75.90% HM versus 71.79% for full fine-tuning, demonstrating that asymmetrically adapting only the 3D encoder while freezing shared semantic encoders prevents cross-modal drift and improves domain shift robustness. These highlight ReFine3D's potential for scalable and generalizable adaptation of 3D foundation models.

# 5    Conclusion

In this paper, we propose ReFine3D, a regularized fine-tuning framework for 3D multi-modal foundation models. Our method tackles two key challenges: overlooking the unique characteristics of 3D point clouds and under-utilization of pre-trained VLMs' visual semantics. ReFine3D addresses these issues through selective layer tuning, consistency-based regularization, and test-time augmentation with confidence-based aggregation strategies. Furthermore, ReFine3D uses the pre-trained VLMs visual knowledge by aligning point cloud-text-vision triplets in a shared representation space. We demonstrate its effectiveness across multiple 3D domain generalization benchmarks, showing consistent gains in task-specific performance and cross-domain robustness.

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
