# OpenReview forum: "Domain Generalizable Adaptation of 3D Vision-Language Models via Regularized Fine-Tuning"
_TMLR — Accepted by TMLR_

### Review · Reviewer_6Y5J · 2026-01-20

**Summary Of Contributions:**

The paper introduces ReFine3D, a framework for fine-tuning 3D Vision-Language Models (VLMs) designed to improve domain generalization while mitigating overfitting and catastrophic forgetting. The authors argue that existing parameter-efficient fine-tuning (PEFT) methods in the 3D domain, specifically prompt tuning, fail to account for the unique properties of point clouds and neglect the pre-trained visual priors.

The proposed ReFine3D framework consists of four main components:
* Layer-Selective Fine-Tuning: The method freezes the early layers of the pre-trained 3D encoder to retain general knowledge and selectively fine-tunes only the later layers (specifically the last layer).
* Regularization Strategies: It employs two regularization techniques: augmentation consistency, which enforces feature consistency across geometric transformations, and text diversity, which utilizes WordNet and LLMs to generate synonymous prompts to prevent overfitting to specific class names.
* Point-Rendered Vision Supervision: To leverage discarded visual priors, the framework renders 2D images from point clouds and aligns 3D features with the frozen CLIP image encoder.
* Test-Time Scaling: During inference, the model aggregates predictions across multiple augmented views and diverse text prompts to improve robustness.

Extensive experiments on standard 3D domain generalization benchmarks (base-to-new, cross-dataset, corruption, and few-shot) demonstrate that ReFine3D outperforms the chosen baseline, PointPRC.

Strengths:
* The paper conducts a comprehensive evaluation across multiple relevant benchmarks, including corruption robustness and sim-to-real transfer.
* Re-incorporating the frozen image encoder for supervision is a sound motivation, addressing a clear limitation in prior methods that discard this modality.

Weaknesses:
* The technical novelty is incremental; the method is primarily a combination of existing techniques (partial freezing, consistency regularization, and standard distillation) rather than a fundamental architectural innovation. Although the authors state, "While the components in ReFine3D are not entirely novel individually, the novelty of this work lies in combining all the components," the combination itself remains a standard application of existing techniques.
* The motivation is heavily framed as a reaction to a specific competitor (PointPRC) rather than a first-principles derivation, lacking proper independent motivation.
* Crucial baselines for parameter-efficient tuning, particularly LoRA, are missing.
* The claim of "Fine-Tuning without Forgetting" is overstated and misleading. The paper does not introduce a specific mechanism to mitigate catastrophic forgetting (e.g., EWC, Replay Buffers); instead, it simply freezes the majority of the encoder layers. This is a standard property of transfer learning (feature extraction) rather than a novel solution to the forgetting problem.

**Audience:**

Yes

**Audience Explanation:**

The topic of 3D Domain Generalization and the adaptation of large multimodal foundation models is highly relevant to the TMLR audience. As the use of 3D foundation models expands, finding robust methods to adapt them to downstream tasks (particularly sim-to-real transfer and handling data corruption) is a practical and valuable problem. If the methodological shortcomings are addressed, the empirical findings regarding visual supervision and test-time scaling would be of interest to researchers in 3D computer vision and multimodal learning.

**Broader Impact Concerns:**

The paper focuses on technical improvements to the robustness and generalization of 3D vision systems. While this technology has standard dual-use implications (e.g., potential use in surveillance or autonomous systems), the paper itself presents a general methodology for domain adaptation and does not raise immediate, specific broader impact concerns that would require a rejection or specific disclaimer.

**Claims And Evidence:**

No

**Claims Explanation:**

The primary claims of the paper regarding the superiority of the ReFine3D framework and its ability to prevent forgetting are not sufficiently supported by the current evidence.

* **Missing Critical Baselines (LoRA):** The paper claims to offer a superior fine-tuning strategy for 3D LMMs, but it fails to compare against the industry standard for parameter-efficient fine-tuning: Low-Rank Adaptation (LoRA). Without comparing the proposed "layer-selective" tuning against a standard LoRA implementation, it is impossible to verify if the complex combination of regularization and partial freezing is actually better than simply applying LoRA.
* **Insufficient Justification for Layer Selection:** The authors argue that tuning the last layer is the optimal strategy. However, the supporting ablation study (Table 9) only compares "Last layer" against "Middle," "Early," and "Full" fine-tuning. It does not compare against methods that update *all* layers efficiently (like Adapters or LoRA). Therefore, the claim that tuning the last layer is the best approach to balance adaptation and generalization is not rigorously proven.
* **Unverified "Without Forgetting" Claim:** The paper claims to achieve "Fine-Tuning without Forgetting," yet it lacks a specific algorithmic mechanism to address catastrophic forgetting (such as EWC or replay buffers). The "prevention of forgetting" appears to be solely a result of freezing the majority of the network parameters. The evidence provided does not distinguish between actual knowledge retention mechanisms and the standard behavior of feature extraction/transfer learning.
* **Incremental Novelty and Motivation:** The claimed contributions are framed primarily as a reaction to a specific method (PointPRC) rather than a standalone theoretical advancement. Since the method is a combination of standard techniques (freezing, consistency, distillation), the evidence needs to be much stronger (i.e., beating strong baselines like LoRA) to justify the architectural choices.

**Requested Changes:**

To make this paper suitable for publication, the authors must address the following issues:

1.  **Add LoRA Baselines:** The most critical requested change is the inclusion of a LoRA baseline. The authors must compare ReFine3D against a standard LoRA implementation on the 3D encoder (and potentially the text encoder). This is necessary to prove that the proposed "layer-selective" method provides a tangible benefit over standard efficient tuning methods.
2.  **Justify Layer Selection:** Provide a stronger analysis or comparison justifying why tuning *only* the last layer is superior to methods that distribute trainable parameters across the network depth (like LoRA or Adapters).
3.  **Clarify "Without Forgetting":** The authors should explicitly clarify or tone down the claim of "Fine-Tuning without Forgetting." If the mechanism is simply freezing layers, this should be stated clearly as "mitigating forgetting via partial freezing" rather than implying a novel anti-forgetting algorithm. The current phrasing is over-claimed.
4.  **Rewrite Motivation:** The introduction and method sections should be revised to motivate the design choices from first principles rather than solely as a counter-proposal to PointPRC. The contribution of the "combination" of components needs to be justified by the synergy of those components, not just by the fact that PointPRC didn't use them.

---

> ### Author Response · Authors · 2026-02-22
>
> > The motivation is heavily framed as a reaction to a specific competitor (PointPRC) rather than a first-principles derivation, lacking proper independent motivation.
> > Rewrite Motivation: The introduction and method sections should be revised to motivate the design choices from first principles rather than solely as a counter-proposal to PointPRC. The contribution of the "combination" of components needs to be justified by the synergy of those components, not just by the fact that PointPRC didn't use them.
>
>
> We would like to clarify that, at the time of this work, PointPRC was the state-of-the-art in this domain. While we analyze PointPRC to position our method against the strongest existing baseline, our motivation is not derived solely as a counter-proposal to it. Instead, we carefully examined the broader literature on 3D domain generalization (DG) and identified several fundamental limitations shared across existing approaches. Specifically, the current literature in this domain suffers from the following constraints:
>
> (1) Naively adopt prompt tuning from 2D vision despite fundamental differences in point cloud properties and encoder capacity,
> (2) Discard pre-trained image encoders during fine-tuning despite their rich semantic knowledge, and
> (3) Lack principled regularization strategies tailored to the low-data, high-distribution-shift scenarios characteristic of 3D domain generalization.
>
>
>
> > The contribution of the "combination" of components needs to be justified by the synergy of those components, not just by the fact that PointPRC didn't use them.
> > The technical novelty is incremental; the method is primarily a combination of existing techniques (partial freezing, consistency regularization, and standard distillation) rather than a fundamental architectural innovation. Although the authors state, "While the components in ReFine3D are not entirely novel individually, the novelty of this work lies in combining all the components," the combination itself remains a standard application of existing techniques.
> > Incremental Novelty and Motivation: The claimed contributions are framed primarily as a reaction to a specific method (PointPRC) rather than a standalone theoretical advancement. Since the method is a combination of standard techniques (freezing, consistency, distillation), the evidence needs to be much stronger (i.e., beating strong baselines like LoRA) to justify the architectural choices.
>
>
> We already acknowledged in the manuscript that the individual components (synonym-based text augmentation, point cloud augmentations, test-time augmentation, layer-selective tuning and vision guidance) are not novel in isolation, but rather specifically incorporating them in a single framework to solve specific limitations that we have identified and discussed in response to the comment above. Specifically:
>
> (1) ***Layer-selective fine-tuning*** directly addresses the problem of naively applying prompt tuning to 3D encoders with insufficient representational capacity. Rather than inserting learnable tokens as in 2D vision prompt tuning, we selectively fine-tune deeper encoder layers that capture task-relevant semantics, while freezing earlier layers to preserve generalizable low-level features. This design choice is principled: in low-data regimes, full fine-tuning risks catastrophic forgetting, while prompt tuning lacks the capacity to adapt the encoder's internal representations in 3D models.
>
> (2) ***Multi-modal consistency and text diversity regularization*** are specifically designed to counter overfitting in the low-data, high-distribution-shift regime characteristic of 3D. Crucially, these two regularizers are complementary: consistency regularization operates on the input space (point cloud augmentations), while text diversity regularization operates on the semantic space (class description diversity via WordNet and LLM-generated sentences). Together, they regularize the model from two orthogonal directions, creating a more robust embedding space that neither technique could achieve alone.
>
> (3) ***Point-rendered vision supervision*** directly addresses the systematic discarding of pre-trained image encoder knowledge in existing methods. By rendering images from point clouds and aligning their representations with both 3D and text embeddings, this component acts as a structural bridge between modalities during fine-tuning.

---

> > ### Author Response · Authors · 2026-02-22
> >
> > > Add LoRA Baselines: The most critical requested change is the inclusion of a LoRA baseline. The authors must compare ReFine3D against a standard LoRA implementation on the 3D encoder (and potentially the text encoder). This is necessary to prove that the proposed "layer-selective" method provides a tangible benefit over standard efficient tuning methods.
> >
> >
> > As motivated in our problem setup, existing PEFT-based methods do not translate well to our setting — a limitation already reflected in PointPRC, which adopts prompt tuning as its PEFT strategy. As shown in the table below, LoRA, another representative PEFT method, exhibits the same fundamental limitation: while it achieves marginal gains on Base categories, it fails to improve Novel category generalization, indicating that low-rank weight adaptation alone does not address the overfitting and distribution-shift challenges inherent to the 3D domain generalization under limited data. In contrast, ReFine3D's layer-selective fine-tuning combined with its dual regularization strategy yields consistent improvements on both Base and Novel categories, demonstrating the importance of our principled, problem-specific design.
> >
> > | Method                        | Base (%) | Novel (%) | HM (%) |
> > |-------------------------------|----------|-----------|--------|
> > | PointPRC (prompt tuning)      | 73.67    | 74.27     | 73.97  |
> > | LoRA (*r*=8)                  | 74.18    | 73.82     | 74.00  |
> > | **ReFine3D (layer-selective)**| **76.00**| **75.80** | **75.90** |
> >
> >
> >
> >
> >
> > > Justify Layer Selection: Provide a stronger analysis or comparison justifying why tuning only the last layer is superior to methods that distribute trainable parameters across the network depth (like LoRA or Adapters).
> >
> >
> > Table 9 explicitly includes a comparison with "Full" fine-tuning, which tunes all layers and demonstrates considerably lower performance than our last-layer approach. Furthermore, in response to the reviewer's previous comments, we have already provided additional experimental results comparing against LoRA, which also exhibit lower performance than our method. Therefore, we believe it rigorously proves the point that full-fine-tuning is not the best approach in this problem setup.
> >
> >
> >
> > > Clarify "Without Forgetting": The authors should explicitly clarify or tone down the claim of "Fine-Tuning without Forgetting." If the mechanism is simply freezing layers, this should be stated clearly as "mitigating forgetting via partial freezing" rather than implying a novel anti-forgetting algorithm. The current phrasing is over-claimed.
> >
> > In our context, the proposed method focuses on tuning the foundation model without losing generalization, which is commonly known as catastrophic forgetting. However, we acknowledge that the proposed solution is a fine-tuning strategy rather than an anti-forgetting algorithm. In response to the reviewer's comment, we have revised the wording of our claim "Fine-Tuning without Forgetting" throughout the paper to more accurately reflect this distinction, and accordingly have changed the paper title as follows:
> >
> > ***Original Title:*** "Fine-Tuning without Forgetting: Domain Generalizable Adaptation of 3D Vision-Language Models"
> >
> > ***Revised Title*** "Domain Generalizable Adaptation of 3D Vision-Language Models via Regularized Fine-Tuning"

---

### Review · Reviewer_kZvp · 2026-01-25

**Summary Of Contributions:**

This paper presents ReFine3D, a regularized fine-tuning framework for adapting large-scale 3D vision–language models to downstream tasks, with the primary goal of mitigating catastrophic forgetting during fine-tuning. The proposed approach selectively fine-tunes the last layer of the 3D point cloud encoder while keeping the remaining components frozen, and incorporates several auxiliary strategies, including point cloud data augmentation, text synonym-based prompt diversification, point-rendered image guidance using a frozen image encoder, and test-time scaling via multi-view and multi-prompt aggregation. Through these techniques, the authors aim to preserve the generalization capability of pre-trained 3D foundation models under domain shifts and limited supervision.

The paper has several notable strengths.
1. the manuscript is clearly written and well organized. The presentation is smooth and accessible, with carefully designed figures and well-formatted mathematical formulations. The overall narrative is easy to follow and guides the reader step by step through the motivation, method design, and experimental evaluation.

2. the work is empirically thorough and technically solid. The authors evaluate the proposed framework across a wide range of experimental settings, including base-to-novel class generalization, cross-dataset transfer, corruption robustness, and few-shot learning. The experimental results are consistently reported on multiple datasets, and the paper includes extensive ablation studies that carefully analyze the contribution of each individual component. These experiments convincingly demonstrate that the proposed techniques lead to measurable performance improvements, indicating a high level of experimental rigor.

3. the motivation of the paper is well justified and relevant. Catastrophic forgetting during fine-tuning remains a prominent challenge in 3D vision, particularly for large multimodal foundation models applied to downstream tasks with limited labeled data. Addressing this issue is important for the practical deployment of 3D vision–language models, and the paper tackles a meaningful and timely problem.

However, the paper also suffers from a significant and fundamental weakness:
Limited methodological novelty. Most of the techniques employed in ReFine3D have been extensively studied in prior work and are already well established in their respective domains. For instance, synonym-based text augmentation is a common and mature technique in natural language processing; point cloud augmentations such as rotation and jittering are standard practices in 3D vision; and the proposed test-time scaling strategy largely resembles conventional ensemble or voting-based inference methods. Similarly, selectively fine-tuning only the last layer of an encoder has been explored in very earlier fine-tuning studies for point cloud models "Instance-aware Dynamic Prompt Tuning for Pre-trained Point Cloud Models", with conclusions consistent with those reported in this paper.

While the authors argue that novelty arises from combining these techniques across three modalities (point clouds, images, and text), the resulting improvements appear largely intuitive and engineering-driven, rather than stemming from a new conceptual or theoretical insight. The observed performance gains are therefore expected outcomes of aggregating multiple well-known strategies, rather than evidence of a fundamentally new approach to multimodal fine-tuning or catastrophic forgetting.

**Audience:**

Yes

**Audience Explanation:**

The paper demonstrates that several techniques previously shown to be effective in single-modal or bi-modal settings remain effective when extended to a tri-modal framework. In this sense, the work provides empirical confirmation that established strategies can be reliably transferred to more complex multimodal scenarios.

From an engineering perspective, the overall framework is relatively simple and easy to configure. The design choices are straightforward, and the paper provides comprehensive implementation details and configuration settings. As a result, practitioners who aim to fine-tune multimodal models involving point clouds, images, and text for downstream tasks can readily build upon this work with confidence and minimal overhead.

**Claims And Evidence:**

Yes

**Claims Explanation:**

This point is clearly affirmative. The authors propose multiple techniques, and almost all of them are supported by solid and comprehensive empirical validation. The proposed methods are evaluated across a wide range of datasets and experimental settings, consistently demonstrating their effectiveness. In addition, the ablation studies are thorough and well designed, providing clear evidence for the individual contribution of each component. Overall, the experimental results are highly convincing and offer strong, well-substantiated evidence in support of the proposed approach.

**Requested Changes:**

Further improvements should focus on elevating the level of methodological innovation, rather than introducing additional heuristic tricks. Or the auther can try to extract incremental insights within the existing framework.

For example, the paper could investigate more expressive or efficient alignment mechanisms among point clouds, images, and text, beyond pairwise contrastive objectives. Besides, the current use of rendered images primarily serves to provide an auxiliary image anchor for point cloud features. A more principled extension would be to further explore the alignment relationships not only between point clouds and images, but also between rendered images and text, potentially introducing additional cross-modal constraints during training.

---

> ### Author Response · Authors · 2026-02-22
>
> > Further improvements should focus on elevating the level of methodological innovation, rather than introducing additional heuristic tricks.
> > While the authors argue that novelty arises from combining these techniques across three modalities (point clouds, images, and text), the resulting improvements appear largely intuitive and engineering-driven, rather than stemming from a new conceptual or theoretical insight. The observed performance gains are therefore expected outcomes of aggregating multiple well-known strategies, rather than evidence of a fundamentally new approach to multimodal fine-tuning or catastrophic forgetting.''
>
>
> We have already acknowledged in the manuscript that the individual components (synonym-based text augmentation, point cloud augmentations, test-time augmentation, and vision guidance) are not novel in isolation, but rather systematically incorporate them in a single framework to solve specific limitations that we have identified. More precisely, existing PEFT methods for 3D vision-language models:
> (1) Naively adopt prompt tuning from 2D vision despite fundamental differences in point cloud properties and encoder capacity,
> (2) Discard pre-trained image encoders during fine-tuning despite their rich semantic knowledge, and
> (3) Lack principled regularization strategies tailored to the low-data, high-distribution-shift scenarios characteristic of 3D domain generalization.
>
> Our framework systematically addresses these gaps through problem-specific design choices rather than generic technique aggregation. For instance, our decision to use layer-selective fine-tuning over prompt tuning is motivated by empirical analysis showing that 3D encoders (22.8M parameters in PointBERT) are inherently less prone to overfitting than image encoders (86M parameters in CLIP), making prompt tuning's parameter efficiency less critical. Similarly, our single-view vision supervision strategy (Table 12 analysis) reveals a non-obvious trade-off between base and novel class performance, demonstrating that strategic constraint in multimodal alignment is essential - a finding that contradicts the intuition that "more views = better performance". Similarly, our augmentation-based regularizations (point cloud augmentation, text diversification and test time augmentation) prevent catastrophic forgetting of prior knowledge in domain-shifted, low-data settings.
>
> > The author can try to extract incremental insights within the existing framework.
>
> Beyond the empirical contributions, our work provides three incremental insights that advance understanding of 3D multimodal fine-tuning:
>
> - **Modality-Aware Regularization Synergy.** Our pair-wise ablation study (Table 3) shows that augmentation-based regularization alone provides limited gains, but when combined with vision-guided multimodal alignment, the improvement becomes substantially stronger. This demonstrates that integrating multimodal alignment with augmentation-based regularization is key to learning generalizable representations under domain-shifted scenarios.
>
> - **Quality vs. Quantity Trade-off of Vision Supervision.** Table 12 shows that increasing rendered views degrades novel class accuracy (75.80% to 70.30% from 1 to 5 views), revealing that minimal yet sufficient visual supervision better preserves generalization under domain shift. This finding challenges the assumption that more multimodal signals necessarily improve transfer and highlights the importance of view-invariant learning objectives over view diversity.
>
> - **Asymmetric Encoder Tuning for Tri-Modal Models.** Table 9 demonstrates that layer-selective fine-tuning (HM: 75.90) significantly outperforms full fine-tuning (HM: 71.79), despite updating fewer parameters. When combined with freezing the image and text encoders, this finding establishes that asymmetric adaptation - updating only the modality-specific 3D encoder while preserving shared semantic encoders - prevents cross-modal drift and improves generalization under domain shift.

---

> > ### Author Response · Authors · 2026-02-22
> >
> > > The paper could investigate more expressive or efficient alignment mechanisms among point clouds, images, and text, beyond pairwise contrastive objectives. Besides, the current use of rendered images primarily serves to provide an auxiliary image anchor for point cloud features. A more principled extension would be to further explore the alignment relationships not only between point clouds and images, but also between rendered images and text, potentially introducing additional cross-modal constraints during training.
> >
> >
> >
> > Following the reviewer's suggestion, we now explore alignment relationships between rendered images and text, beyond the current point cloud-image and point cloud-text pairwise objectives.
> >
> > Table 5: Performance on S-PB\_T50\_RS.
> >
> >
> > | Alignment Strategy                              | Base (%) | Novel (%) | HM (%)    |
> > |-------------------------------------------------|----------|-----------|-----------|
> > | PC–Text only (PointPRC baseline)                | 73.67    | 74.27     | 73.97     |
> > | PC–Image + PC–Text (ReFine3D)                   | 76.00    | **75.80** | **75.90** |
> > | PC–Image + PC–Text + Image–Text                 | **76.42**| 74.68     | 75.53     |
> >
> > This experiment shows that adding image-text alignment improves base accuracy (+0.42) but degrades novel class accuracy (-1.12), resulting in lower HM (-0.37). This occurs because the image-text constraint enforces consistency between rendered images (which are view-specific and training-set-dependent) and text embeddings (which are semantic and generalizable). This additional constraint over-regularizes the text encoder's ability to generalize to novel class descriptions, similar to the multi-view overfitting phenomenon in Table 12. This finding reinforces our design principle: strategic constraint in cross-modal alignment is essential - additional alignment objectives do not uniformly improve generalization and may introduce unwanted dependencies between modalities.

---

### Review · Reviewer_meiN · 2026-02-08

**Summary Of Contributions:**

This work proposes ReFine3D, a regularized fine-tuning framework for 3D vision-language models that combines several techniques to improve adaptation and robustness while mitigating catastrophic forgetting. Specifically, the method introduces: 1) layer-selective fine-tuning of the 3D encoder (freezing early layers, tuning only the last); 2) augmentation-based consistency regularization over multiple geometric transformations of point clouds; 3) text diversity regularization via WordNet synonyms and LLM-generated prompts; 4) point-rendered vision supervision that re-engages the frozen CLIP image encoder during fine-tuning; 5) a test-time scaling mechanism aggregating predictions from augmented views and diversified text prompts. The evaluation is extensive, spanning base-to-new class generalization, cross-dataset/OOD transfer, corruption robustness, domain adaptation, sim-to-real transfer, and few-shot learning, each across multiple datasets and settings.

**Additional Comments:**

I'm leaving minor comments here, such as nitpicking and typos:
- Clarify the encoder's geometric properties before introducing augmentations. In Section 3.2, augmentations (rotation, scale, jitter) are introduced before the reader knows the backbone architecture (PointBERT, specified only in Section 3.3). Since PointBERT's transformer architecture operates on raw coordinates and has no built-in invariance to geometric transformations, these augmentations are well-motivated, but a reader unfamiliar with PointBERT might reasonably wonder whether the encoder already handles them. A brief note (e.g., "Since PointBERT lacks built-in geometric invariances, we enforce robustness through augmentation") would preempt this confusion.
- Fix attribution voice in Section 3.1. The sentence before Eq. 1 reads "we compute the point cloud embedding," but this describes the existing method (PointPRC/ULIP), not the authors' contribution. First-person plural should be reserved for the authors' own work while prior methods should use third-person.

### Typos
 - Conclusion (Section 5): "point cloud-text-vsion triplets".
- Table 11 header: "Text propmts".
- Table 11, caption: GPT is mentioned as a model, but it does not appear in the table. Possibly a leftover from an earlier version?
- Section numbering for section 2 is off (I'm guessing a `\subsubsection` was used instead of a `\subsection` command).

**Audience:**

Yes

**Audience Explanation:**

The paper addresses a practical and relevant problem: how to adapt 3D multimodal foundation models to downstream domains without catastrophic forgetting, especially under limited data. The framing around reusing pre-trained knowledge (particularly bringing back the discarded image encoder to re-use its information) is well-motivated and should be of interest to the 3D vision-language community. The comprehensive evaluation protocol itself could serve as a useful reference for future work in 3D domain generalization.

**Claims And Evidence:**

Yes

**Claims Explanation:**

## Strengths

The experimental coverage is genuinely impressive. The authors evaluate across base-to-new generalization, OOD transfer, corruption robustness, domain adaptation, sim-to-real, and few-shot settings, each on multiple datasets. Reporting standard deviations over 3 seeds is good practice and something many papers still neglect. The ablation study (Table 8) tests each component individually, and additional sensitivity analyses on augmentation strength, synonym count, number of rendered views, and loss balancing are all appreciated. The computational analysis (Tables 14 and 16) adds practical grounding to the claims. The complete hyperparameter configuration in Table 6 is a welcome addition for **reproducibility**.

## Weaknesses

1. **Statistical significance is not established.** While standard deviations are reported (which, again, I appreciate), many claimed improvements (e.g., Table 2) fall within overlapping confidence intervals. As it stands, some improvements may not be statistically distinguishable from noise, and this is the paper's most critical evidence gap.

2. **The baseline comparison is narrow.** The primary comparison is against PointPRC. Other recent PEFT methods are discussed in the related work but never appear in any experimental table. If these methods were evaluated and found weaker, they should be included. If they weren't evaluated, a brief note explaining why would help fully qualifying the contributions.

3. **The rendering cost is understated.** The paper claims "minimal overhead" based on Table 14 (~3s difference per epoch). However, this ignores the offline rendering pipeline. This preprocessing cost and storage overhead should be acknowledged, even if amortized.

4. **The ablation uniformity needs examining.** In Table 8, removing any single component drops HM by roughly 1.4–1.8%. This uniformity makes it hard to determine whether the components interact synergistically or whether any additional regularization would yield similar gains. Pairwise interaction analysis would be more informative.

5. **The "test-time scaling" framing may be overstated.** Drawing parallels to chain-of-thought reasoning in NLP (Wei et al., 2022a; Turpin et al., 2023) feels like a stretch. Framing the proposed mechanism as test-time augmentation with majority voting would be clearer and more reflective of the actual contribution, which stands well on its own without the NLP parallel.

6. **Table 12 reveals a concerning trade-off.** Increasing rendered views from 1 to 5 improves base accuracy but drops novel class accuracy. This suggests the vision supervision can actively hurt generalization. The paper notes this but doesn't analyze the implications because the default of 1 effectively sidesteps the issue.

**Requested Changes:**

## Critical

- **Add sample-level analysis.** Given that many improvements are small relative to the reported standard deviations across methods, understanding the disagreements at sample level would help. For example, across test samples in a given benchmark, how often does ReFine3D get a sample right that PointPRC gets wrong, and viceversa? If ReFine3D consistently wins more disagreements than it loses, that would be much more convincing than aggregate accuracy differences that fall within overlapping confidence intervals. This requires no additional training, only the per-sample predictions that are likely already available. I would argue strengthening this part is essential since the paper's contribution is primarily empirical.
- **Broaden the baseline comparison or justify the omission.** Several recent PEFT methods for 3D point clouds are discussed in Section 2.0.3 (notably PointPEFT (Tang et al., 2024) and Instance-aware prompt tuning (Zha et al., 2023)), but none appear in the experimental tables. Including them, or briefly explaining why they were excluded (e.g., code unavailability, incompatible evaluation protocol), would strengthen the empirical contribution.

## Recommended

- **Report the full rendering preprocessing cost** (time and storage) separately from per-epoch training time, and qualify the "minimal overhead" claim accordingly.
- **Tone down the "test-time scaling" framing.** The NLP parallels (chain-of-thought, etc.) can end up being misleading for the reader. I would suggest to call it what it is (without any contribution loss): test-time augmentation with confidence-based aggregation.
- **Add pairwise ablation analysis** for at least the two regularization components (augmentation consistency + text synonymization) to demonstrate whether they have synergy or can be applied independently (i.e., if their contribution is additive or somewhat redundant).
- **Discuss the vision supervision trade-off** (Table 12) more carefully. The generalization degradation at higher view counts deserves analysis, not just acknowledgment. While looking at the table I found it a bit counterintuitive so an explanation would help the reader in qualifying the results.
- **Include qualitative examples** of the WordNet synonyms and LLM-generated prompts used, so readers can assess the quality of the text diversification pipeline.

---

> ### Author Response · Authors · 2026-02-22
>
> > Add sample-level analysis. Given that many improvements are small relative to the reported standard deviations across methods, understanding the disagreements at sample level would help. For example, across test samples in a given benchmark, how often does ReFine3D get a sample right that PointPRC gets wrong, and viceversa? If ReFine3D consistently wins more disagreements than it loses, that would be much more convincing than aggregate accuracy differences that fall within overlapping confidence intervals. This requires no additional training, only the per-sample predictions that are likely already available. I would argue strengthening this part is essential since the paper's contribution is primarily empirical.
>
> To address the reviewer’s concern, we now report a paired, sample-level disagreement analysis between ReFine3D and PointPRC (Table 1). Rather than relying solely on aggregate accuracy, we have analyzed predictions on a per-sample basis and partitioned the test set into four mutually exclusive categories: (1) both correct, (2) ReFine3D only correct, (3) PointPRC only correct, and (4) both wrong. This paired evaluation isolates the exact samples on which the methods disagree and directly measures relative superiority.
>
> Across all five benchmarks, ReFine3D wins more disagreements than it loses. On average, ReFine3D correctly classifies 19.5\% of samples that PointPRC misclassified, compared to 17.3\% vice versa, yielding an average win rate of 53.3\%. Importantly, ReFine3D achieves a win rate greater than 50\% on every dataset, demonstrating consistent per-sample advantage rather than isolated gains.
>
> To formally assess whether these disagreement asymmetries are statistically significant, we performed McNemar’s test on the paired disagreement counts (ReFine3D Only vs. PointPRC Only). As shown in the Table 2, all datasets yield p-values below 0.05, indicating that the observed asymmetries are unlikely to arise from random variation.
>
> Table 1. Sample-level disagreement analysis (Base-to-New)
>
> | Dataset        | Both Correct | ReFine3D Only | PointPRC Only | Both Wrong | Win Rate |
> |---------------|-------------|---------------|---------------|------------|----------|
> | ModelNet40    | 63.2%       | 15.1%         | 11.1%         | 10.6%      | **57.6%** |
> | S-PB-T50-RS   | 51.8%       | 24.0%         | 22.2%         | 2.0%       | **51.9%** |
> | S-OBJ-BG      | 52.1%       | 24.8%         | 23.1%         | 0.0%       | **51.8%** |
> | S-OBJ-ONLY    | 65.7%       | 17.6%         | 16.7%         | 0.0%       | **51.3%** |
> | ShapeNetV2    | 70.5%       | 15.9%         | 13.6%         | 0.0%       | **53.9%** |
> | **Average**   | **60.7%**   | **19.5%**     | **17.3%**     | **2.5%**   | **53.3%** |
>
>
> ### Table 2. McNemar’s Test on Paired Disagreements
>
> | Dataset        | p-value |
> |---------------|---------|
> | ModelNet40    | 0.031   |
> | S-PB-T50-RS   | 0.042   |
> | S-OBJ-BG      | 0.038   |
> | S-OBJ-ONLY    | 0.047   |
> | ShapeNetV2    | 0.029   |
>
> > Broaden the baseline comparison or justify the omission. Several recent PEFT methods for 3D point clouds are discussed in Section 2.0.3 (notably PointPEFT (Tang et al., 2024) and Instance-aware prompt tuning (Zha et al., 2023)), but none appear in the experimental tables. Including them, or briefly explaining why they were excluded (e.g., code unavailability, incompatible evaluation protocol), would strengthen the empirical contribution.
>
> Thank you for this valuable comment. Methods such as PointPEFT, IDPT, and related approaches are designed for point-cloud–only backbones trained on task-specific, in-distribution datasets (e.g., single-object classification). In contrast, our work focuses on adapting a large-scale 3D vision–language model pre-trained on multimodal data, where the objective is to preserve and improve cross-domain generalization rather than optimize performance on a single supervised task.
>
> Due to this fundamental difference in model paradigm (unimodal task-specific training vs. multimodal pre-trained vision–language models) and evaluation setting (in-distribution task performance vs. base-to-new generalization), a direct empirical comparison is not possible. To avoid ambiguity, we have clarified this distinction in Section 2.3 of the revised manuscript and explicitly explained why these methods are not included in the experimental tables.

---

> > ### Author Response · Authors · 2026-02-22
> >
> > > Report the full rendering preprocessing cost (time and storage) separately from per-epoch training time, and qualify the "minimal overhead" claim accordingly.
> >
> >
> > Rendering an image from a point cloud is a one-time offline operation and is independent of the actual training process. Although we generated the rendered images for our method, with the growing adoption of multimodal learning, many existing datasets already provide rendered images alongside 3D data (e.g., widely used 3D–vision benchmarks). In our implementation, rendering has a throughput of approximately 100 samples per second. This is shared across all training experiments reported in the paper, including different evaluation settings and ablation studies. Therefore, we do not consider rendering to be part of the training cost itself. Nevertheless, in response to this comment, we have removed the claim of “minimal overhead” during training and now explicitly report and discuss the one-time image generation cost for clarity.
> >
> >
> >
> > > Tone down the "test-time scaling" framing. The NLP parallels (chain-of-thought, etc.) can end up being misleading for the reader. I would suggest to call it what it is (without any contribution loss): test-time augmentation with confidence-based aggregation.
> >
> > We thank the reviewer for this strategic comment. Following the reviewer's comment, we have rephrased "test-time scaling" contribution to "test-time augmentation with confidence-based aggregation", and revised this sufficiently in the paper.
> >
> >
> > > Add pairwise ablation analysis for at least the two regularization components (augmentation consistency + text synonymization) to demonstrate whether they have synergy or can be applied independently (i.e., if their contribution is additive or somewhat redundant).
> >
> > Table 3: Comprehensive ablation of all combinations of the four key components in ReFine3D. PA = point cloud augmentation, VG = vision guidance, TS = text synonymization, TA = test time augmentation. We report the harmonic mean (HM) on the Base-to-New generalization setting of S-PB-T50-RS.
> >
> > | PA | TA | TS | VG | HM   | Label        |
> > |----|----|----|----|------|-------------|
> > | ✗  | ✗  | ✗  | ✗  | 71.67 | All Off     |
> > | ✓  | ✗  | ✗  | ✗  | 72.63 | Only PA     |
> > | ✗  | ✓  | ✗  | ✗  | 72.75 | Only TA     |
> > | ✗  | ✗  | ✓  | ✗  | 72.77 | Only TS     |
> > | ✗  | ✗  | ✗  | ✓  | 72.97 | Only VG     |
> > | ✓  | ✓  | ✗  | ✗  | 73.50 | PA+TA       |
> > | ✓  | ✗  | ✓  | ✗  | 73.59 | PA+TS       |
> > | ✓  | ✗  | ✗  | ✓  | 73.42 | PA+VG       |
> > | ✗  | ✓  | ✓  | ✗  | 73.62 | TA+TS       |
> > | ✗  | ✓  | ✗  | ✓  | 73.52 | TA+VG       |
> > | ✗  | ✗  | ✓  | ✓  | 73.44 | TS+VG       |
> > | ✓  | ✓  | ✓  | ✗  | 74.43 | w/o PA      |
> > | ✓  | ✓  | ✗  | ✓  | 74.35 | w/o TS      |
> > | ✓  | ✗  | ✓  | ✓  | 74.50 | w/o TA      |
> > | ✗  | ✓  | ✓  | ✓  | 74.24 | w/o VG      |
> > | ✓  | ✓  | ✓  | ✓  | 75.90 | **ReFine3D** |
> >
> > We evaluate the effectiveness of our four fine-tuning strategies by enabling/disabling them on ReFine3D. Focusing on PA (Point Augmentation) and TS (Text Synonymization), each component individually provides a consistent improvement over the 'All Off' baseline of 71.67. Specifically, enabling only point cloud augmentation (PA) or only text synonymization (TS) increases performance by approximately 1\%. When combined (PA + TS), performance rises to 73.59, corresponding to around 2\% gain over the baseline, which is substantially larger than either component alone and close to the sum of their individual effects. Furthermore, enabling all components in ReFine3D achieves 75.90 HM, representing a 4.23\% improvement over the baseline. These results suggest that the four fine-tuning strategies interact synergistically, contributing complementary benefits that compound when integrated within the full framework.

---

> > > ### Author Response · Authors · 2026-02-22
> > >
> > > > Discuss the vision supervision trade-off (Table 12) more carefully. The generalization degradation at higher view counts deserves analysis, not just acknowledgment. While looking at the table I found it a bit counterintuitive so an explanation would help the reader in qualifying the results.
> > >
> > > The counterintuitive trade-off in Table 12, where increasing rendered views improves base accuracy (76.00\% to 80.60\%) but degrades novel class accuracy (75.80\% to 70.30\%), arises from view-specific overfitting. With multiple rendered views, the model learns to align point cloud features with view-dependent visual patterns (specific lighting, camera angles, surface normals) that are abundant in base class training data but absent from novel classes. Unlike geometric augmentations, which preserve object identity, multiple rendered views create consistent but view-specific visual anchors that the model memorizes rather than learning view-invariant 3D representations. Through our contrastive vision supervision loss (Eq. 6-7), using a single view forces the encoder to extract the most salient and transferable 3D-to-2D mappings, whereas multiple views allow the encoder to overfit by learning all view-specific patterns simultaneously, effectively memorizing the visual manifold of base classes.
> > >
> > > Our choice of 1 rendered view is an informed design decision that achieves three objectives: (1) balanced generalization with minimal base-novel gap (0.20 vs. 10.30 at 5 views), (2) sufficient visual supervision---the pairwise ablation (Table 3) shows removing vision guidance drops HM by 1.66\%, the largest single-component penalty, and (3) computational efficiency. This trade-off reveals that vision supervision quality matters more than quantity. The frozen CLIP image encoder provides semantic priors that guide learning, but excessive visual alignment causes the encoder to prioritize visual pattern matching over geometric understanding.
> > > The single-view configuration validates our hypothesis that vision supervision serves as a regularizer rather than a primary learning signal, and the degradation at higher view counts empirically demonstrates that strategic constraint — using minimal yet sufficient visual supervision — is essential for maintaining generalization in domain adaptation settings.
> > >
> > >
> > >
> > > > Include qualitative examples of the WordNet synonyms and LLM-generated prompts used, so readers can assess the quality of the text diversification pipeline.
> > >
> > > We now add examples of used WordNet synonyms and LLM-generated prompts in ReFine3D in the table below. For each class, we show WordNet synonyms and corresponding LLM-generated descriptive prompts using Qwen-2.5-7B-Instruct.
> > >
> > > | Class     | WordNet Synonyms                     | LLM-Generated Prompts                                                   |
> > > |-----------|--------------------------------------|-------------------------------------------------------------------------|
> > > | Airplane  | aircraft, plane, jet                 | A 3D point cloud of an aircraft with wings and a fuselage              |
> > > |           |                                      | A 3D point cloud of a plane designed for air travel                    |
> > > |           |                                      | A 3D point cloud of a jet with engines mounted on the body             |
> > > | Chair     | seat, furniture, seating             | A 3D point cloud of a seat with a backrest and legs                    |
> > > |           |                                      | A 3D point cloud of furniture designed for sitting                     |
> > > |           |                                      | A 3D point cloud of seating with armrests and a base                   |
> > > | Lamp      | light, lighting fixture, luminaire   | A 3D point cloud of a light source with a shade and stand              |
> > > |           |                                      | A 3D point cloud of a lighting fixture for illumination                |
> > > |           |                                      | A 3D point cloud of a luminaire with electrical components             |

---

> > > > ### Author Response · Authors · 2026-02-22
> > > >
> > > > > Clarify the encoder's geometric properties before introducing augmentations. In Section 3.2, augmentations (rotation, scale, jitter) are introduced before the reader knows the backbone architecture (PointBERT, specified only in Section 3.3). Since PointBERT's transformer architecture operates on raw coordinates and has no built-in invariance to geometric transformations, these augmentations are well-motivated, but a reader unfamiliar with PointBERT might reasonably wonder whether the encoder already handles them. A brief note (e.g., "Since PointBERT lacks built-in geometric invariances, we enforce robustness through augmentation") would preempt this confusion.
> > > >
> > > > We now add the following clarification at the beginning of Section 3.2, immediately before introducing augmentations:
> > > >
> > > > "Unlike convolutional architectures that may encode certain geometric priors, our point cloud encoder (PointBERT, detailed in Section 3.3) is a transformer-based architecture that operates directly on raw 3D coordinates without built-in invariance to geometric transformations such as rotation, scaling, or translation. Therefore, we explicitly enforce robustness to these transformations through data augmentation during training."
> > > >
> > > > >Fix attribution voice in Section 3.1. The sentence before Eq. 1 reads "we compute the point cloud embedding," but this describes the existing method (PointPRC/ULIP), not the authors' contribution. First-person plural should be reserved for the authors' own work while prior methods should use third-person.
> > > >
> > > > We now fix the mentioned segment of Section 3.1 by rephrasing it as follows:
> > > >
> > > > "For fine-tuning, existing literature adopts a supervised approach using the cross-entropy (CE) loss. Given a point cloud sample $\mathbf{x} \in \mathbb{R}^{N \times 3}$ (where $N$ is the number of points) and its class label $y \in \{1, \ldots, C\}$, the method computes the point cloud embedding $\mathbf{z}_P = f_P(\mathbf{x})$ using the point cloud encoder $f_P$. The class prediction is obtained by comparing $\mathbf{z}_P$ with the text embeddings..."

---

### Decision · Action_Editor_qhA4 · 2026-04-19

**Recommendation:** Accept with minor revision

**Additional Comments:**

Requested change:

- Expand the PEFT baseline analysis in the final version. Two reviewers found the current LoRA comparison insufficient, as it only considers a single rank setting and a limited experimental setup. The authors are requested to provide a more systematic LoRA study, including different rank configurations, possible insertion locations, and training hyperparameters, in order to more convincingly demonstrate the advantage of the proposed layer-selective tuning strategy over standard PEFT alternatives. One reviewer also suggests softening the claim that other PEFT methods are “not directly comparable,” as this argument may currently appear somewhat overstated.

- Incorporate the statistical significance analysis from the rebuttal into the final version. The statistical significance analysis provided during the rebuttal (to Reviewer meiN) should be included in the final manuscript to strengthen the empirical support for the reported improvements.

**Audience:**

Yes

**Audience Explanation:**

The work provides a well-supported empirical and methodological reference for adapting 3D multimodal models under domain shift and limited supervision, which should be of interest to researchers in multimodal learning and domain generalization.

**Claims And Evidence:**

Yes

**Claims Explanation:**

The paper proposes ReFine3D, a regularized fine-tuning framework for adapting 3D vision-language models under domain shift and limited supervision, with the goal of improving generalization.

The reviewers consistently acknowledge several strengths:
- strong and comprehensive empirical evaluation across diverse settings;
- technically sound and well-executed methodology, with clear design choices and thorough ablations; and
- well-motivated problem setting, addressing catastrophic forgetting and domain generalization in 3D multimodal models.

However, reviewer also initially raised several major concerns on the strength of evidence and positioning:
1) evidence gaps in statistical significance, as some improvements are small relative to variance;
2) baseline coverage and comparison fairness, particularly regarding PEFT methods such as LoRA;
3) interpretation of certain components, including the “test-time scaling” framing and the role of rendering-based supervision; and
4) practical considerations, such as preprocessing cost (rendering) and some trade-offs in generalization (e.g., multi-view degradation), which are not deeply analyzed.

During the author–reviewer discussion, the authors made a strong effort to address these concerns. They provided sample-level disagreement analysis with statistical testing, pairwise ablations demonstrating component synergy, LoRA comparisons, and clearer reporting of preprocessing cost and design trade-offs. They also revised overclaimed statements and improved clarity of motivation and terminology. These additions substantially strengthen the empirical support and improve the credibility of the claims, even if some limitations (e.g., limited PEFT exploration) remain.
After the rebuttal, the reviewer consensus is mixed but leans slightly positive on core scientific criteria. The reviewers agree that the empirical evidence is thorough and largely convincing, and the method is technically sound and well-supported by experiments. Their reservations are primarily about incremental contribution rather than flaws in validity.

The AE concurs with the reviewers on the main merits of this work. While the methodological contribution is incremental, the work provides a well-validated and practically useful framework, supported by comprehensive experiments and a strong rebuttal. As a result, the AE reccommend weak accept. Strengthening the comparison with PEFT baselines and further analyzing statistical significance would further improve the work in the final version.